# Convergence of two global regulators to coordinate expression of essential virulence determinants of *Mycobacterium tuberculosis*

**Hina Khan[†], Partha Paul, Ritesh Rajesh Sevalkar[‡], Sangita Kachhap[§], Balvinder Singh, Dibyendu Sarkar***

CSIR-Institute of Microbial Technology, Chandigarh, India

**\*For correspondence:**
dibyendu@imtech.res.in

**Present address:** [†]Department of Biosciences and Bioengineering, Indian Institute of Technology Roorkee, Uttarakhand, India; [‡]Department of Microbiology, University of Alabama, Birmingham, United States; [§]Jerzy Haber Institute of Catalysis and Surface Chemistry Polish Academy of Sciences, Niezapominajek, Poland

**Competing interest:** The authors declare that no competing interests exist.

**Abstract** Cyclic AMP (cAMP) is known to function as a global regulator of *Mycobacterium tuberculosis* gene expression. Sequence-based transcriptomic profiling identified the mycobacterial regulon controlled by the cAMP receptor protein, CRP. In this study, we identified a new subset of CRP-associated genes including virulence determinants which are also under the control of a major regulator, PhoP. Our results suggest that PhoP as a DNA binding transcription factor, impacts expression of these genes, and phosphorylated PhoP promotes CRP recruitment at the target promoters. Further, we uncover a distinct regulatory mechanism showing that activation of these genes requires direct recruitment of both PhoP and CRP at their target promoters. The most fundamental biological insight is derived from the inhibition of CRP binding at the regulatory regions in a PhoP-deleted strain owing to CRP-PhoP protein-protein interactions. Based on these results, a model is proposed suggesting how CRP and PhoP function as co-activators of the essential pathogenic determinants. Taken together, these results uncover a novel mode of regulation where a complex of two interacting virulence factors impact expression of virulence determinants. These results have significant implications on TB pathogenesis.

## Editor's evaluation

This paper will be of broad interest to those working on the regulation of gene expression and mycobacteria. The article presents data to elucidate the collaboration between two important transcriptional regulators to selectively turn on the expression of a gene set in *Mycobacterium tuberculosis*.

## Introduction

*Mycobacterium tuberculosis*, as one of the most successful human pathogens, retains the ability to adapt to diverse intracellular and extracellular environment it encounters during infection, persistence, and transmission (*Barry et al., 2009*). Thus, *M. tuberculosis* is able to survive within interior of macrophages, and droplet nuclei in the atmosphere that are generated by individuals infected with the bacilli. While inhalation of droplets causes the disease spread, following infection *M. tuberculosis* can persist in a non-replicating (dormant) state for a long period of time. A characteristic feature of tuberculosis is that the disease actually occurs through reactivation of dormant infection under favorable conditions (e.g., under compromised immunity of infected individuals). In keeping with this, about a third of the world's population is believed to have latent *M. tuberculosis* infection (*Dye et al., 1999*) with a lifetime risk of reactivation of TB as high as 5–10% (*Bloom and Murray, 1992*). Therefore,

appropriate regulation of gene expression in mycobacteria is considered critical for establishing and emerging from the dormant state.

More than 100 transcription regulators, 11 two-component systems, 6 serine-threonine protein kinases, and 13 alternative sigma factors are present in *M. tuberculosis*, suggesting that the complex transcriptional regulation is important for mycobacterial pathogenesis. Thus, to understand in vivo physiology of *M. tuberculosis,* it is becoming increasingly important to investigate the role of individual regulators and their participation in integrated networks. In the recent past, cyclic AMP (cAMP) has been shown to function as a global regulator of mycobacterial gene expression with a critical involvement in host-pathogen interactions and virulence (*Agarwal et al., 2006*; *Gazdik and McDonough, 2005*; *Rickman et al., 2005*). Little is yet known about the regulation of cAMP-associated mycobacterial genes.

The classical model of bacterial cAMP regulation is best studied in *Escherichia coli* (*Busby and Ebright, 1999*), where cAMP signal is transduced by CRP. Upon cAMP binding, CRP undergoes a conformational change, shows an enhanced recognition to specific DNA sequence, and plays a central role to coordinate global transcriptional regulation for optimal bacterial utilization of different carbon sources. High throughput ChIP-sequencing data suggest that *E. coli* CRP shows extensive low-affinity binding, suggesting a possible role of the regulator as a chromosome organizer (*Grainger et al., 2005*). The corresponding *M. tuberculosis* CRP (encoded by Rv3676) (*Agarwal et al., 2006*; *Bai et al., 2005*; *Rickman et al., 2005*), which shares a sequence identity of 32% and similarity of 53% over 189 residues of *E. coli* CRP (*Rickman et al., 2005*), recognizes a similar DNA binding motif and is shown to regulate transcription of a large number of mycobacterial genes (*Rickman et al., 2005*). Although cAMP binds to *M. tuberculosis* CRP with weaker affinity and lesser impact on DNA-binding (*Bai et al., 2007*), polymorphisms in CRP which result in enhanced DNA binding have been reported to influence transcription of a number of genes for *Mycobacterium bovis* BCG (*Hunt et al., 2008*; *Spreadbury et al., 2005*). Consistent with its global role in transcription regulation, a CRP-deleted mutant is implicated in virulence because of its significant growth attenuation in mice and macrophages (*Rickman et al., 2005*). Further, CRP shows the highest activation of *rpfA* and *whiBI* genes (*Rickman et al., 2005*), encoding proteins that are involved in reviving dormant bacteria (*Mukamolova et al., 2002*) and whiBI encodes wbI family proteins (*den Hengst and Buttner, 2008*; *Soliveri et al., 2000*) that function to control developmental processes. Thus, CRP is implicated in controlling mycobacterial developmental processes that regulates persistence and/or emergence from the dormant state. However, mechanisms of transcription activation of CRP-regulated mycobacterial genes in response to metabolic signals remain poorly understood.

While probing genome-wide transcriptomic profile coupled with ChIP-seq/SELEX data of major regulators, we observed that a subset of genes is under the positive regulation of two major regulators, CRP (*Kahramanoglou et al., 2014*) and PhoP (*Galagan et al., 2013*; *He and Wang, 2014*; *Solans et al., 2014*; *Walters et al., 2006*). The PhoP protein, the response regulator of the PhoPR two-component system (TCS) (*Gupta et al., 2006*), belongs to the PhoB/OmpR subfamily of transcription factors, characterized by an N-terminal regulatory domain and a C-terminal DNA binding domain (*Pathak et al., 2010*). Inactivation of *phoP* (*ΔphoP*-H37Rv) significantly reduces the multiplication of the bacilli in ex vivo and in vivo infection models (*Walters et al., 2006*), suggesting that PhoPR remains essential for virulence (*Pérez et al., 2001*). Further, *ΔphoP*-H37Rv reveals a significantly lowered synthesis of cell-wall components diacyltrehaloses, polyacyltrehaloses, and sulfolipids, specific to pathogenic mycobacterial species, relative to wild-type *M. tuberculosis* (WT-H37Rv) (*Gonzalo Asensio et al., 2006*; *Goyal et al., 2011*; *Walters et al., 2006*), suggesting an additional mechanism of attenuation. Notably, the attenuation due to deletion of *phoPR* is so significant that deletion strains are currently being considered in trials as a vaccine strain (*Arbues et al., 2013*). Along the line, a single-nucleotide polymorphism (S219L) in the DNA binding domain of PhoP affects ESX-1 gene expression system leading to failure of ESX-1 -dependent ESAT-6 secretion, absence of which accounts for the loss of virulence of the attenuated strain *M. tuberculosis* H37Ra (*Frigui et al., 2008*; *Lee et al., 2008*). In an attempt to define PhoP regulon, microarray-based transcriptomic studies and high throughput RNA-sequencing analyses uncovered that approximately 2% of the H37Rv genome is regulated by PhoP (*Solans et al., 2014*; *Walters et al., 2006*). These include genes belonging to lipid and intermediary metabolism, PE, PPE, and PE_PRGS families and transcriptional regulator(s) categories (*Gonzalo Asensio et al., 2006*; *Walters et al., 2006*). To probe how PhoP functions as a

DNA binding transcription factor, previous studies have identified the consensus DNA binding of the regulator within its target promoters (*Goyal et al., 2011*; *He and Wang, 2014*). More recent studies highlight role of the *phoP* locus in numerous stress response of mycobacteria (*Abramovitch et al., 2011*; *Baker et al., 2019*; *Baker et al., 2014*; *Bansal et al., 2017*; *Goar et al., 2022*; *Sevalkar et al., 2019*; *Singh et al., 2014*). Despite this body of knowledge, we are yet to identify the signal that activates the PhoPR system and transcriptional regulators acting downstream of PhoP.

In this study, we sought to define the scope of individual regulators and their participation in integrated regulatory networks. Here, we uncover a novel role of PhoP in the mechanism of CRP-dependent mycobacterial gene expression. While these results account for the failure to activate a few CRP-regulated genes in Δ*phoP*-H37Rv, we provide evidence showing that an activation of a subset of genes requires the simultaneous presence of both CRP and PhoP at the target promoters. Insightfully, results reported in this work account for an explanation of how CRP-dependent activation of a subset of mycobacterial genes is also linked to positive regulation by PhoP (*Solans et al., 2014*; *Walters et al., 2006*) underscoring a critical role of PhoP in virulence gene expression via a complex mechanism of transcriptional control involving functional cooperation of the two virulence regulators.

## Results

### *phoP* directly regulates a subset of genes of CRP regulon

DNA microarray and sequence-based transcriptomic profiling identified PhoP regulon (*Solans et al., 2014*; *Walters et al., 2006*). The role of the virulence regulator PhoP in cAMP-responsive mycobacterial gene expression became apparent when we noted an overlap between the regulons under the control of cAMP receptor protein (CRP) and PhoP. Thus, we compared relative expression of representative CRP regulon genes in WT bacilli (referred to as WT-H37Rv) and a mutant strain of *M. tuberculosis* H37Rv, in which *phoPR* locus (Rv0757-Rv0758) has been inactivated (referred to as Δ*phoP*-H37Rv), under normal, acidic pH and exposure to NO conditions (*Figure 1A*). Notably, these two stress conditions are among the major hostile conditions that *M. tuberculosis* encounters within the host (*Nathan and Shiloh, 2000*; *Rustad et al., 2009*; *Wayne and Sohaskey, 2001*). Our results demonstrate that a subset of CRP-regulated genes including *icl1*, *umaA*, and *whiB1* are positively regulated by PhoP. We also compared the expression of these genes in the mutant complemented with wild-type *phoP* or *phoPD71N* allele, a mutant PhoP defective for phosphorylation at Asp-71 (*Gupta et al., 2006*). In keeping with phosphorylation-dependent activation of PhoP, PhoPD71N was unable to restore gene expression. Notably, *icl1* has been implicated in persistence and virulence of *M. tuberculosis* in macrophages and mice (*Fontán et al., 2008*; *Gonzalo-Asensio et al., 2008*; *Gupta et al., 2009*; *Menon and Wang, 2011*; *Rohde et al., 2007*; *Schnappinger et al., 2003*), whereas *umaA* encodes for a mycolic acid synthase (*Laval et al., 2008*). On the other hand, *whiB1* as an essential gene has been shown to function in NO sensing of mycobacteria (*Smith et al., 2010*). Since we noted that elevated expression of these CRP-regulated genes in the complemented mutant (Δ*phoP::phoP*), we compared *phoP* expression in WT-H37Rv and complemented strain (*Figure 1—figure supplement 1*). Our results demonstrate that *phoP* expression level is reproducibly higher in the complemented mutant (relative to the WT bacilli) both under normal conditions (*Figure 1—figure supplement 1A*) as well as during growth under low pH conditions (*Figure 1—figure supplement 1B*). These results possibly account for elevated mRNA levels of a few representative genes in the complemented mutant relative to WT-H37Rv (*Figure 1A*). However, relatively poor restoration of gene expression in the complemented mutant under acid stress is possibly related to inadequate activation of PhoPR under specific stress conditions.

Because the above genes are controlled by *M. tuberculosis* CRP, we examined if PhoP impacts expression of these genes by controlling *crp* expression (*Figure 1B*). Our results show that *crp* is not under the regulation of *phoP* locus. However, low pH-inducible *aprA* which belongs to the PhoP regulon, showed a significantly lowered expression in Δ*phoP*-H37Rv relative to WT-H37Rv, suggesting that PhoP-dependent regulation of cAMP-inducible/CRP-controlled genes is not attributable to PhoP controlling CRP expression. To investigate whether PhoP is directly recruited within these promoters, Flag-tagged PhoP was ectopically expressed in WT-H37Rv and ChIP-qPCR was carried out using anti-Flag antibody (*Figure 1C*). Notably, PhoP is significantly enriched at the *whiB1*, *icl1*, and *umaA* promoters by 12.4±0.4-fold, 4.4±0.5-fold, and 6.2±0.9-fold, respectively, relative to mock sample (no

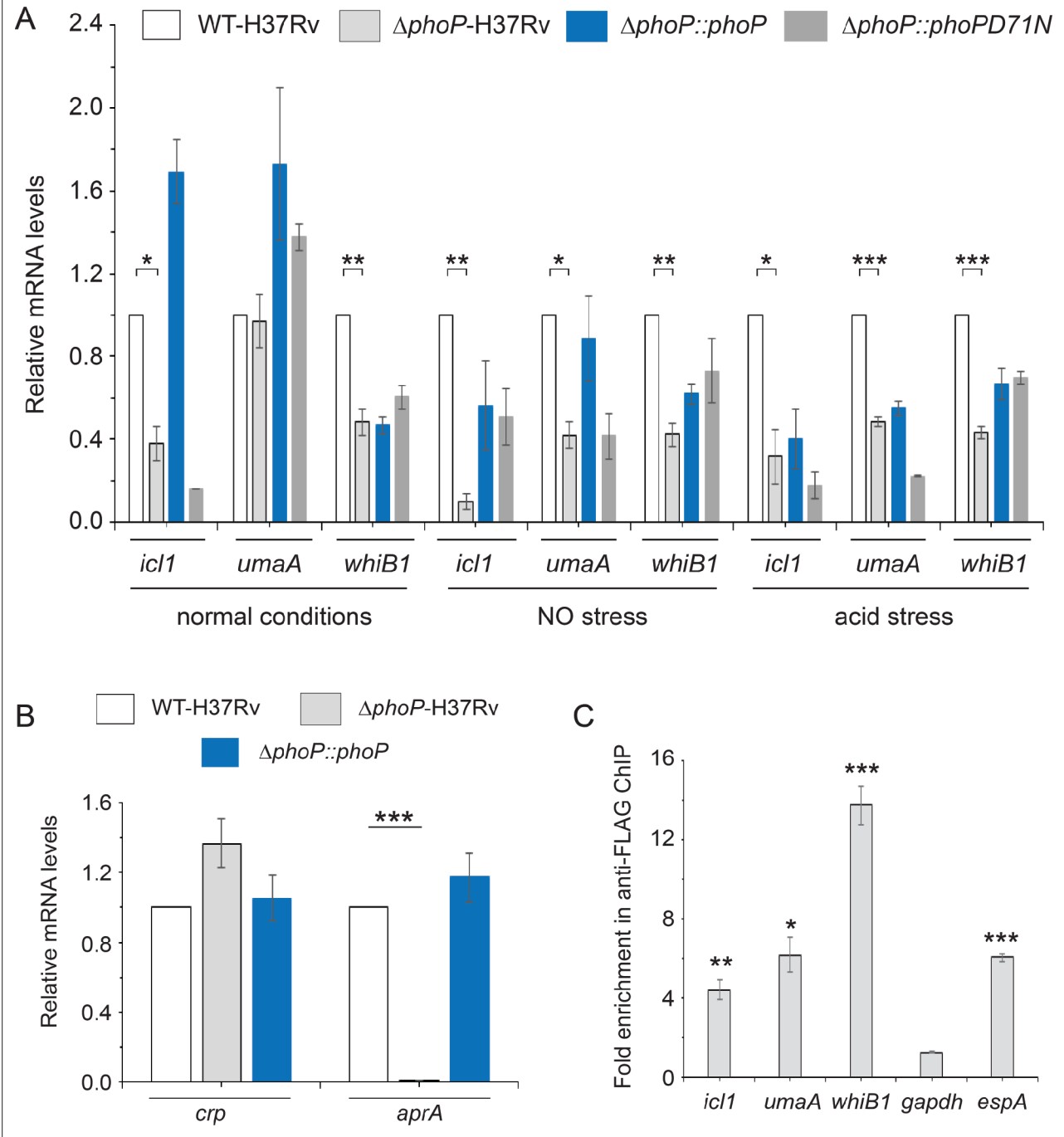

**Figure 1.** PhoP-depletion impacts expression of a subset of CRP-regulon genes. (**A**) Quantitative RT-PCR was carried out to compare the expression level of *icl1*, *umaA*, and *whiB1* in indicated mycobacterial strains, grown under normal and specific stress conditions. Note that mycobacterial strains include WT-H37Rv (empty bar), Δ*phoP*-H37Rv (gray bar), and the mutant strain complemented with wild-type *phoP* (blue bar) or phosphorylation-defective *phoPD71N* (dark gray bar). The results show average values from biological triplicates, each with two technical repeats (*p≤0.05; **p≤0.01; ***p≤0.001). Fold changes in mRNA levels were determined as described in the Materials and methods. Notably, changes in gene expression levels were insignificant when compared between Δ*phoP*-H37Rv (gray bar) and the mutant complemented with phosphorylation-defective *phoPD71N* (dark gray bar). (**B**) To examine role of PhoP in *crp* expression, RT-qPCR compared expression levels of *crp* (Rv3676) in WT-H37Rv (empty bar), Δ*phoP*-H37Rv (gray bar) and the complemented mutant (black bar) (***p≤0.001). PhoP-dependent *aprA* expression was shown as a control. (**C**) In vivo recruitment of PhoP within target promoters was examined by ChIP-qPCR as described in the Materials and methods. *espA* promoter (espAup), and *gapdh*-specific enrichments were used as a positive and negative control, respectively. The experiments were performed in biological duplicates, each with two technical repeats (**p≤0.01; ***p≤0.001), and fold enrichment was determined relative to an IP sample without adding antibody (mock control). Non-significant differences are not indicated.

*Figure 1 continued on next page*

*Figure 1 continued*

The online version of this article includes the following figure supplement(s) for figure 1:

**Figure supplement 1.** *phoP* expression in WT-H37Rv and complemented *ΔphoP*-H37Rv under normal and acidic conditions of growth.

antibody control). However, *gapdh*-specific qPCR did not show any enrichment, whereas a previously reported PhoP-regulated *espA* promoter (**Anil Kumar et al., 2016**) showed 6.0±0.3-fold enrichment, confirming specific recruitment of PhoP within the CRP-regulated promoters.

## Identifying core binding site of PhoP at the CRP-regulated *whiB1* promoter

Next, to identify the core binding sites of PhoP EMSA experiments were carried out using end-labeled whiB1$^{up1}$ (−266 to +60 with respect to the ORF start site of whiB1) and purified PhoP (**Figure 2**). Note that in ChIP assay upstream regulatory region of *whiB1* showed the most efficient in vivo recruitment of PhoP (**Figure 1C**). In keeping with the ChIP data, *P*~PhoP showed ≈20-fold more efficient DNA binding to whiB1$^{up}$ compared to the unphosphorylated regulator (compare lanes 2–4 and lanes 5–7, **Figure 2A** [based on limits of detection in these assays]). Additional experiments using PCR-amplified overlapping promoter fragments show that although 111 bp whiB1$^{up1}$ (−266 to −156 with respect to the ORF start site of whiB1) efficiently binds to *P*~PhoP (lanes 2–3), two other fragments whiB1$^{up2}$

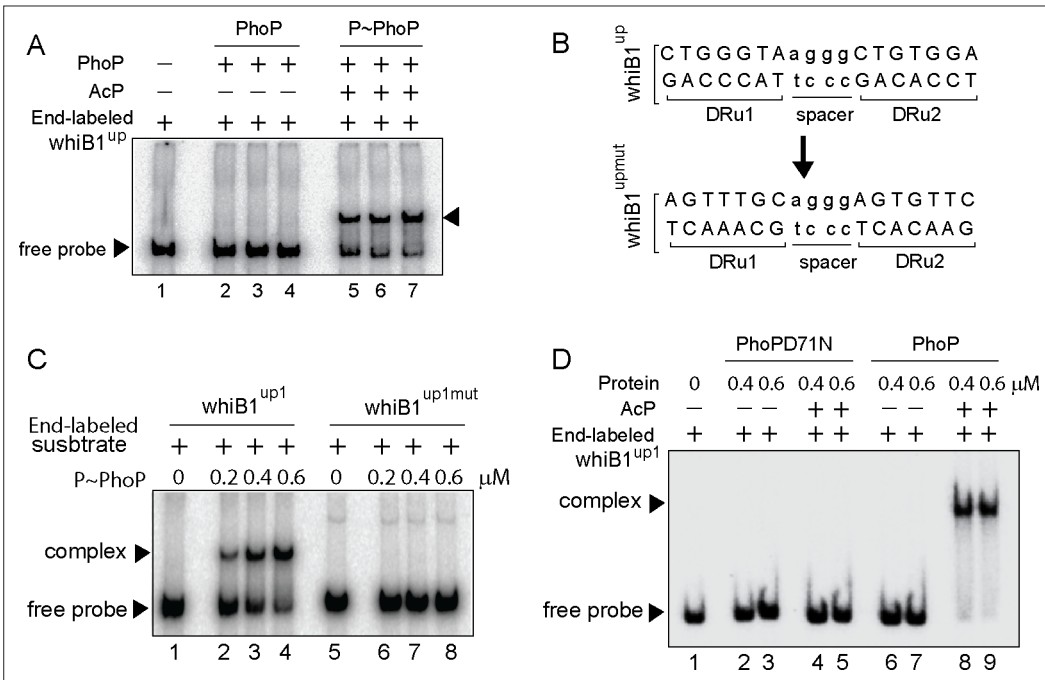

**Figure 2.** Probing core PhoP binding site within *whiB1* promoter region. (**A**) EMSA of radio-labeled whiB1$^{up}$ for binding of 0.2, 0.4, and 0.6 μM of PhoP (lanes 2–4) or *P*~PhoP (lanes 5–7), pre-incubated in a phosphorylation mix with acetyl phosphate (AcP) as the phospho-donor. The arrowheads on the left and right indicate the free probe and a slower moving complex, respectively. (**B**) PhoP binding motif consists of upstream (DRu1) and downstream (DRu2) repeat units. To construct mutant promoter (whiB1upmut), changes in both the repeat units were introduced by changing As to Cs and Gs to Ts and vice versa, and the orientation of DRu2 was reversed. whiB1upmut represents whiB1up fragment carrying changes only at the PhoP binding site. (**C**) EMSA experiment of labeled whiB1up (lanes 2–4), and whiB1upmut (lanes 6–8) to increasing concentrations of *P*~PhoP. The free probe and the slower moving complexes are indicated on the figure. (**D**) EMSA of radio-labeled whiB1up1 for binding of increasing concentrations of phosphorylation-deficient PhoPD71N (lanes 2–5) or PhoP (lanes 6–9), pre-incubated in phosphorylation mixture with or without AcP, respectively. Lane 1 shows the free probe. The assay conditions, sample analyses, and detection of radio-active samples are described in the Materials and methods.

The online version of this article includes the following figure supplement(s) for figure 2:

**Figure supplement 1.** Probing core PhoP binding site within *whiB1* promoter region.

(−156 to −46), and whiB1$^{up3}$ (−50 to +60) failed to form a stable complex with P~PhoP (lanes 5–6, and lanes 8–9, respectively), indicating absence of PhoP binding site downstream of −156 nucleotide of the whiB1 promoter (*Figure 2—figure supplement 1A*). Importantly, while presence of 10-fold and 20-fold molar excess of unlabeled whiB1$^{up1}$ efficiently competed out PhoP binding (lanes 3–4, *Figure 2—figure supplement 1B*), an identical fold excess of unlabeled whiB1$^{up3}$ resulted in a minor variation of PhoP binding (<5%; lanes 5–6) to whiB1$^{up1}$ compared to 'no competitor' control (lane 2). The sequence-specific DNA binding is consistent with the presence of two neighboring PhoP boxes (*He and Wang, 2014*; *Solans et al., 2014*) comprising a 7-bp direct repeat motif spanning nucleotides −97 to −91, and −86 to −80 relative to the whiB1 transcription start site. To examine the importance of this motif, mutations were introduced within both the repeat units of whiB1$^{up1}$ (*Figure 2B*). Notably, a labeled DNA fragment carrying only these changes (whiB1$^{up1mut}$) failed to form a stable PhoP—promoter DNA complex, while the WT promoter (whiB1$^{up1}$) under identical conditions, exhibited efficient DNA binding (compare lanes 2–4 with lanes 6–8, *Figure 2C*). These results suggest that PhoP most likely binds to whiB1$^{up1}$ by sequence-specific recognition of the 7 bp direct repeat motif. Further, our EMSA data comparing DNA binding by PhoP and PhoPD71N reveals that the mutant remains ineffective for DNA binding to radio-labeled whiB1$^{up1}$ both in absence or presence of AcP; however, wild-type PhoP upon pre-incubation with AcP, under identical conditions, showed effective DNA binding (compare lanes 2–5 with lanes 6–9, *Figure 2D*).

Next, whiB1$^{upmut}$ was cloned into a promoter-less mycobacterial reporter plasmid (*Dussurget et al., 1999*) and used as a transcription fusion in *M. smegmatis*. The strain harboring transcriptional fusions were co-transformed with *phoP* expressing plasmid, and grown in 7H9 medium in the absence or presence of 50 ng/ml anhydrotetracycline (ATc), an inducer of PhoP expression, as described previously (*Goyal et al., 2011*). Consistent with *phoP*-dependent in vivo activation of whiB1 (*Figure 1A*), with induction of PhoP the whiB1$^{up}$-lacZ fusion was significantly activated by 8.8±0.2-fold at 24-hr time point as measured by β-galactosidase activity (*Figure 2—figure supplement 1C*). However, upon induction of PhoP expression cells carrying whiB1$^{upmut}$-lacZ fusion showed a comparable enzyme activity of 0.95±0.03-fold relative to uninduced cells. Note that we observed a comparable induction of PhoP expression in *M. smegmatis* cells carrying either the WT or the mutant promoter (inset to *Figure 2—figure supplement 1C*). Thus, we conclude that the newly identified PhoP-binding site, located upstream of CRP binding site, is necessary and sufficient for PhoP-dependent activation of whiB1. Along the line, based on the knowledge of consensus PhoP binding sequence (*He et al., 2016*; *Solans et al., 2014*), we were able to map likely binding sites of the regulator within the *icl1* and *umaA* promoters (*Figure 2—figure supplement 1D*).

## PhoP promotes CRP recruitment at the *whiB1* promoter

Having discovered PhoP binding site in the close vicinity of previously identified CRP binding site (*Kahramanoglou et al., 2014*; *Smith et al., 2010*), we sought to investigate how a representative promoter DNA accommodates both the regulators. In subsequent binding assays, CRP was unable to bind end-labeled whiB1$^{up1}$ to form a complex stable to gel electrophoresis (lanes 2–3, *Figure 3A*). However, under identical conditions, PhoP showed effective binding to whiB1up1 (lanes 8–9). Interestingly, incubation of both PhoP and CRP together at comparable concentrations showed a striking stimulation of ≈5- to 10-fold of DNA binding leading to formation of a slowest moving band (lanes 4–7). These results demonstrate that the presence of PhoP significantly stimulates CRP recruitment within the *whiB1* promoter. To determine the composition, protein components of the slowest moving complex (lane 5) were isolated, resolved on a Tricine SDS-PAGE and identified by Western blotting using anti-His antibody (lane 2, *Figure 3B*). Purified regulators were resolved alongside to confirm relative migration of the regulators (*Figure 3C*). Our results confirm that the slowest moving band represents a PhoP/CRP/whiB1$^{up1}$ ternary complex.

With the results showing two regulators displaying a cooperative DNA binding, we explored the possibility of CRP-PhoP protein-protein interaction using mycobacterial protein fragment complementation (referred to as M-PFC) assay (*Figure 3D*; *Singh et al., 2006*). In this assay, two interacting mycobacterial proteins are independently fused to the domains of mDHFR (murine dihydrofolate reductase), which upon reconstitution confers bacterial resistance to trimethoprim (TRIM). The bait and prey plasmids were constructed as C-terminal fusions with complementary fragments of mDHFR. While PhoP was expressed from integrative plasmid, *crp* and *phoR* encoding ORFs were expressed

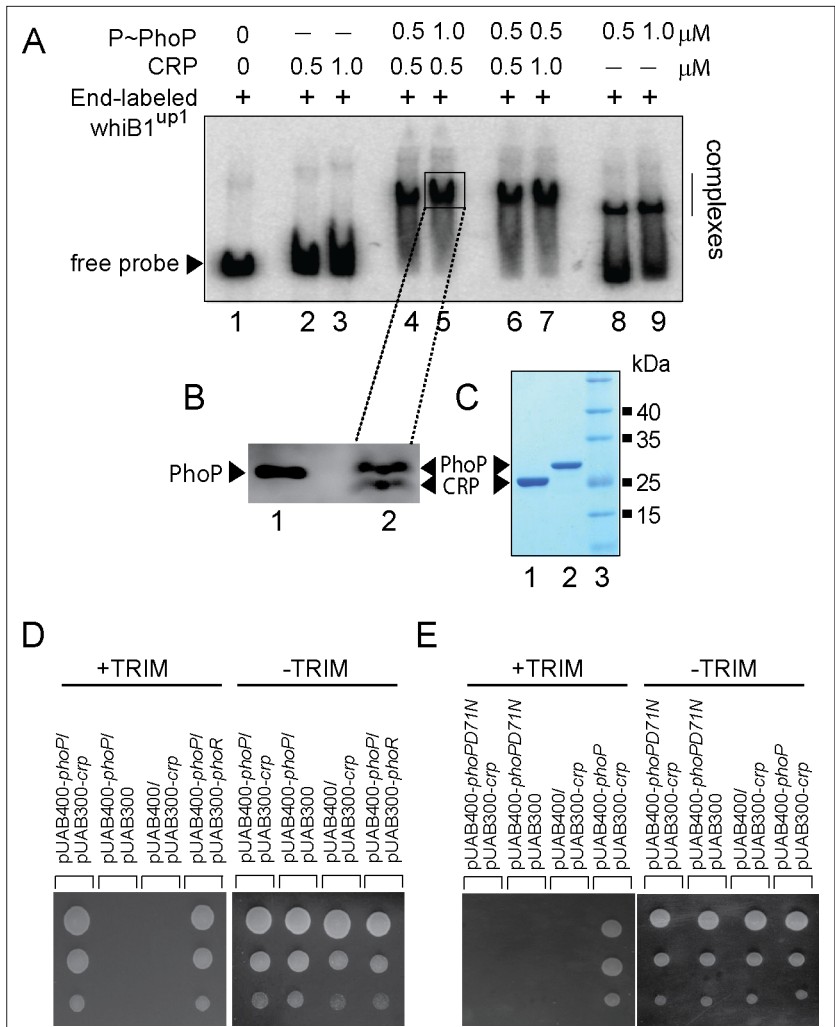

**Figure 3.** PhoP promotes CRP recruitment at the *whiB1* promoter. (**A**) To investigate how the promoter simultaneously accommodates both the regulators, EMSA experiments compared end-labeled whiB1up1 binding to increasing concentrations of purified CRP (lanes 2–3), PhoP (lanes 8–9), and both PhoP and CRP together (lanes 4–7). The assay conditions, sample analyses, and detection are described in the Materials and methods. Positions of the free probe and the complex are indicated on the figure. (**B**) Western blot analyses of protein fraction extracted from the excised gel fragment representing the complex (lane 5), as indicated by a box, was probed by anti-His antibody in lane 2; lane 1 resolved purified PhoP as a control. (**C**) Tricine SDS-PAGE analysis was carried out alongside indicating recombinant His-tagged CRP (lane 1) and PhoP (lane 2); lane 3 resolved marker proteins of indicated molecular masses. Protein samples were visualized by Coomassie blue staining. (**D, E**) To probe protein-protein interaction by M-PFC assays, *Mycobacterium smegmatis* expressing either (**D**) *M. tuberculosis* CRP and PhoP or (**E**) CRP and PhoPD71N, were grown on 7H10/hyg/kan plates in presence of TRIM, and growth was examined for strains co-expressing indicated fusion constructs. In both cases, empty vectors were included as negative controls, and co-expression of pUAB400-*phoP*/pUAB300-*phoR* encoding PhoP and PhoR, respectively, was used as a positive control. All the strains grew well in absence of TRIM.

The online version of this article includes the following figure supplement(s) for figure 3:

**Figure supplement 1.** Probing PhoP-CMR interactions.

from the episomal plasmid. The corresponding plasmids were co-transformed in *M. smegmatis* and transformants were selected on 7H10/kan/hyg plates. Interestingly, cells co-expressing PhoP and CRP grew well in the presence of TRIM (*Figure 3D*), suggesting CRP-PhoP protein-protein interaction. In contrast, *M. smegmatis* harboring empty vector controls showed no detectable growth on 7H10/TRIM plates while these strains grew well on 7H10 plates lacking TRIM.

We next performed M-PFC experiment using phosphorylation-defective PhoPD71N (*Gupta et al., 2006*) and CRP (*Figure 3E*). We found that PhoPD71N is unable to interact with CRP, underscoring the importance of phosphorylation of PhoP on CRP-PhoP interaction. Also, under identical conditions, *M. smegmatis* co-expressing PhoP and CMR, cAMP macrophage regulator which regulates a subset of mycobacterial genes in response to variations of cAMP level during macrophage infection (*Gazdik et al., 2009*), did not show any detectable growth (*Figure 3—figure supplement 1A*). Our RT-qPCR measurements to compare the expression of *M. tuberculosis cmr* and *crp* suggest that the two regulators are expressed in *M. smegmatis* at a comparable level (*Figure 3—figure supplement 1B*), ruling out the possibility of compromised expression of CMR accounting for lack of bacterial growth. Thus, M-PFC data strongly suggest specific interaction between CRP-PhoP. The oligonucleotides used to clone the constructs, and plasmids used for M-PFC experiments are listed in *Supplementary file 1a and b*, respectively.

## Probing CRP-PhoP interactions

To examine the interaction in vivo, whole-cell lysate of Δ*phoP*-H37Rv expressing His-tagged PhoP was treated with DNaseI and incubated with Ni-NTA as described previously (*Anil Kumar et al., 2016*). As expected, both CRP and PhoP were detectable in the crude lysate (lane 1, *Figure 4A*). More importantly, upon elution of bound proteins, the eluent showed a clear presence of CRP (lane 3), suggesting specific in vivo interaction between CRP and PhoP. As a control, we were unable to detect CRP from the eluent using cell lysates of Δ*phoP*-H37Rv carrying empty vector alone (lane 2). Next, during in vitro pull-down assays recombinant PhoP was expressed as a GST fusion protein, and immobilized to glutathione-Sepharose. Following incubation with DNaseI -treated crude lysate of *E. coli* cells expressing His₆-tagged CRP, the column-bound proteins were eluted by 20 mM reduced glutathione. Importantly, we could detect the presence of both CRP and PhoP in the same eluent fraction by immunoblot analysis (lane 1, *Figure 4B*). However, we were unable to detect CRP with only GST-tag (lane 2) or the resin alone (lane 3), allowing us to conclude that PhoP interacts with CRP. In agreement with the M-PFC data shown in *Figure 3E*, our additional pull-down experiments using CRP and phosphorylation-deficient PhoPD71N (*Figure 4C*) unambiguously

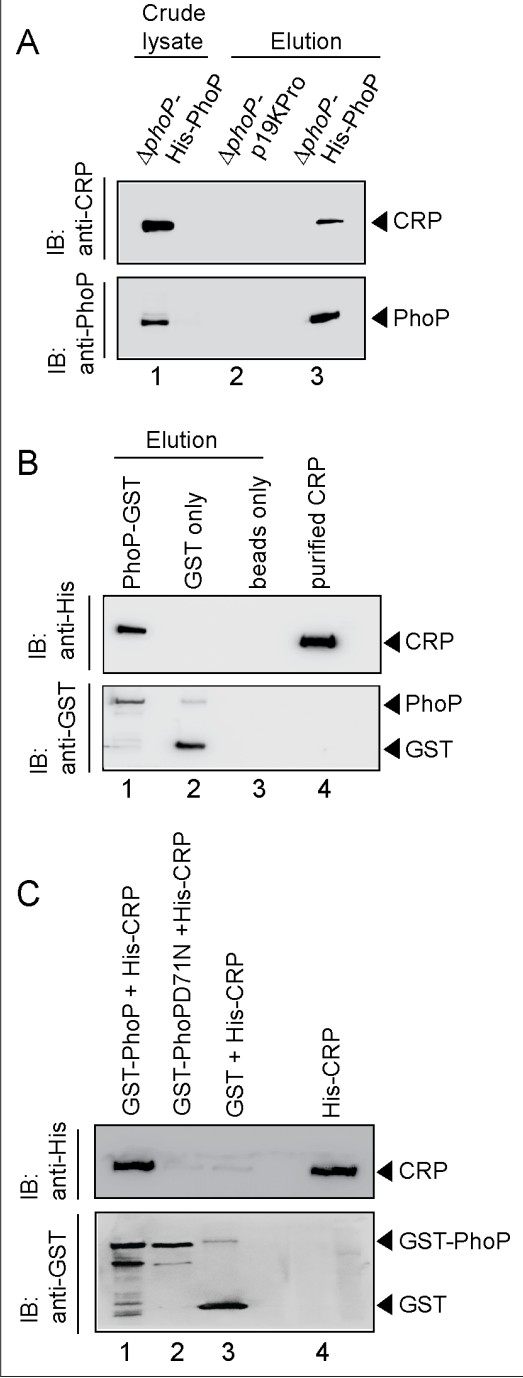

**Figure 4.** Probing CRP-PhoP interactions. (**A**) To examine CRP-PhoP interaction in vivo, DNaseI-treated crude cell lysates of Δ*phoP*-H37Rv expressing His₆-tagged PhoP (p19kpro-*phoP*; *Supplementary file 1b*) was incubated with pre-equilibrated Ni-NTA and eluted with 250 mM imidazole; lane 1, input sample; lane 2, control elution from the crude lysate of cells lacking *phoP* expression; lane 3, co-elution of CRP with PhoP. Blots were probed with anti-PhoP and anti-CRP antibody. (**B**) To investigate CRP-PhoP interaction in vitro, DNaseI-treated crude extract expressing His₆-

*Figure 4 continued on next page*

*Figure 4 continued*

tagged CRP was incubated with glutathione epharose previously immobilized with GST-PhoP. Bound proteins (lane 1) were analyzed by Western blot using anti-His (upper panel) or anti-GST antibody (lower panel). Lane 1 shows presence of CRP bound to GST-PhoP. Identical experiment used glutathione Sepharose immobilized with GST alone (lane 2), or the resin alone (lane 3); lane 4 resolved recombinant $His_6$-tagged CRP. (**C**) To examine whether phosphorylation of PhoP impacts CRP-PhoP interaction, crude lysates of cells expressing $His_6$-tagged CRP was incubated with glutathione-Sepharose previously immobilized with GST-tagged PhoP (lane 1) or PhoPD71N (lane 2), carrying a single substitution of Asp-71 to Asn-71 and therefore, remains ineffective for phosphorylation at Asp-71. Analysis of bound fractions (lanes 1–2) was carried out as described in the legend to **Figure 5** and control sets include glutathione Sepharose immobilized with GST alone (lane 3), or the resin alone (lane 4); lane 5 resolved recombinant $His_6$-tagged CRP.

suggest that phosphorylation of PhoP is necessary for CRP-PhoP interaction.

## CRP and PhoP interact via their corresponding N-terminal domains

Because PhoP was shown to interact with other regulators via its N-terminal domain (**Sevalkar et al., 2019**; **Singh et al., 2020**), we next assessed the role of different stretches of N-terminal domain of PhoP (referred to as $PhoP^N$) in CRP-PhoP interactions. In vitro pull-down assays using mutant GST-PhoP proteins (each with three potential CRP- contacting residues of PhoP replaced with Ala) with His-tagged CRP displayed effective protein-protein interaction as that of the WT PhoP (compare lane 1 with lanes 2–6) (**Figure 5—figure supplement 1A**). These results suggest that either $PhoP^N$ does not contribute to CRP-PhoP interactions or possibly a constellation of residues in a single stretch is involved in CRP-PhoP interaction(s). Next, pull-down assays using His-tagged CRP and a linker deletion mutant of PhoP [PhoPLAla5], replacing five residues spanning $Gly^{142}$ to $Pro^{146}$ with Ala, which remains functional (**Pathak et al., 2010**), suggest that PhoP linker does not appear to contribute to CRP-PhoP interaction(s) (compare lane 1 with lane 2, **Figure 5—figure supplement 1B**). We next probed CRP-PhoP interaction using His-tagged full-length CRP and PhoP domains as GST-fusion constructs (**Figure 5A**). $PhoP^N$ and $PhoP^C$ were previously shown to be functional for phosphorylation and DNA binding activity, respectively (**Pathak et al., 2010**) on their own. Importantly, $PhoP^N$ (comprising PhoP residues 1–141) showed effective protein-protein interaction with CRP (lane 1). However, $PhoP^C$ (comprising PhoP residues 141–247), under identical conditions, did not display an effective interaction with CRP (lane 2). To identify the corresponding interacting domain of CRP, we next used GST-PhoP and His-tagged CRP-domain constructs (**Figure 5B**). Interestingly, $CRP^N$ comprising CRP residues 28–116 co-eluted with GST-PhoP (lane 1). However, $CRP^C$ comprising CRP residues 146–224, under identical conditions, did not co-elute with GST-PhoP (lane 2), suggesting specific interaction between $CRP^N$ and PhoP. Taken together, we surmise that CRP-PhoP interaction is mediated by the N-domains of the corresponding regulators.

Although CRP-PhoP interaction studies were performed in the absence of cAMP, we next undertook in vitro interaction studies using CRP mutants (CRPG79A and CRPT90A), deficient for cAMP binding (**Figure 5—figure supplement 1C**). The mutations were designed based on previously reported CRP structure defining the cAMP binding pocket of the regulator (**Gallagher et al., 2009**). Our pull-down assays demonstrate that the mutant CRP proteins interact with PhoP as effectively as that of the wild-type CRP, allowing us to conclude that cAMP binding to CRP is not required for CRP-PhoP interaction. Although it is conceivable that cAMP binding may induce a conformational change of CRP, in keeping with cAMP-independent functioning of *M. tuberculosis* CRP (**Agarwal et al., 2006**; **Green et al., 2014**; **Stapleton et al., 2010**); the mutant proteins over a range of concentrations showed effective DNA binding to end-labeled $whiB1^{up}$ with a comparable affinity as that of the wild-type CRP (**Figure 5—figure supplement 1D**). These results also suggest that the mutant proteins retain native structure and/or fold as that of wild-type CRP.

## Functioning of CRP requires presence of PhoP

The CRP regulon was previously studied by comparing transcription profiling of WT-H37Rv and a CRP-depleted strain of *M. tuberculosis* H37Rv (Δ*crp*-H37Rv) using RNA-seq and ChIP-seq (**Kahramanoglou et al., 2014**). A careful inspection of ChIP-seq data identifying genome-wide CRP binding sites (**Kahramanoglou et al., 2014**) and SELEX and ChIP-seq data of PhoP (**He et al., 2016**; **Solans et al., 2014**) remarkably uncover the presence of ~10 such promoters comprising CRP and PhoP

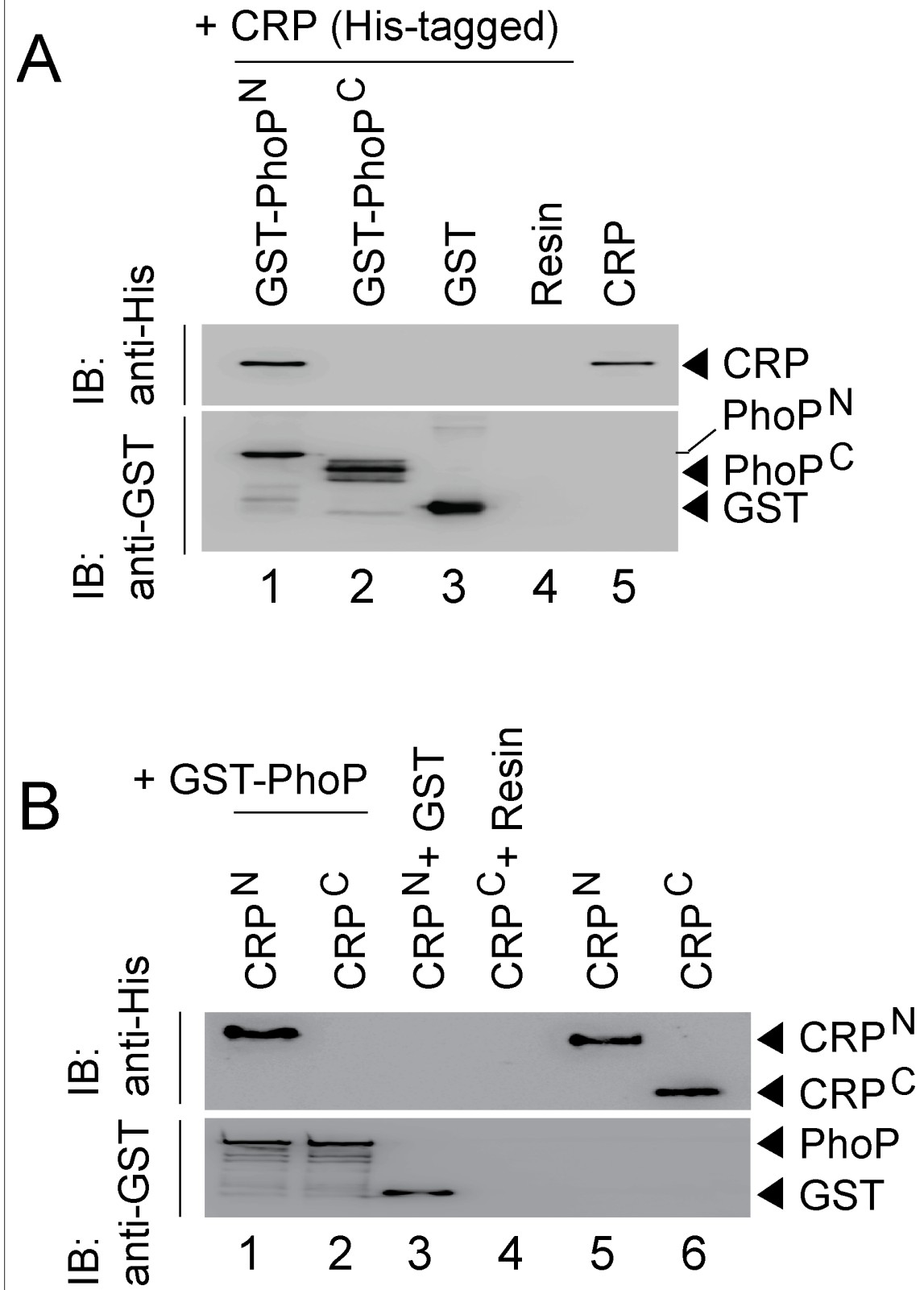

**Figure 5.** CRP and PhoP interact with each other via their corresponding N-terminal domains. (**A, B**) CRP-PhoP interaction was probed by in vitro pull-down assays using either (**A**) His-tagged CRP and GST-tagged PhoP domains (GST-PhoP$^N$ and GST-PhoP$^C$, respectively) or (**B**) GST-tagged PhoP and His-tagged CRP domains (CRP$^N$, and CRP$^C$, respectively). The domain constructs are listed in *Supplementary file 1b*. Fractions of bound proteins were analyzed by Western blot using anti-His (upper panel) or anti-GST antibody (lower panel). Control sets include glutathione Sepharose immobilized with

*Figure 5 continued on next page*

*Figure 5 continued*

GST (lane 3), or the resin (lane 4) alone; lane 5 of (**A**) resolved purified CRP, while lanes 5, and 6 of (**B**) resolved purified CRP$^N$, and CRP$^C$, respectively. The experimental procedures, and data analyses are as described in the legend of *Figure 4B*.

The online version of this article includes the following figure supplement(s) for figure 5:

**Figure supplement 1.** In vitro analysis of CRP-PhoP interactions.

binding sites (*Table 1*). *Figure 6A* shows an overlap of *M. tuberculosis* CRP and PhoP regulon genes that are under direct control of the regulators as determined by the corresponding ChIP-seq data. To examine the influence of PhoP, we next undertook to investigate in vivo recruitment of CRP within its target promoters in WT-and Δ*phoP*-H37Rv. Thus, formaldehyde-cross-linked DNA-protein complexes of growing *M. tuberculosis* cells were sheared to generate the fragments of average size ≈500 bp as described previously (*Singh et al., 2014*). In ChIP experiments using anti-CRP antibody, DNA binding was analyzed by qPCR (*Figure 6B*). Our results show that relative to mock sample, CRP is effectively recruited within its target promoters in WT-H37Rv. For example, *icl1*, and *whiB1* promoters, displayed 5.6±1-fold and 16.4±2.8-fold enrichments of qPCR signals, respectively. In contrast, under identical conditions, Δ*phoP*-H37Rv-derived samples showed insignificant enrichments of CRP within *icl1* (0.7±0.2-fold) and *whiB1* (0.6±0.18-fold), promoters, respectively. However, with identical IP DNA samples, we observed a largely comparable CRP recruitment within the CRP-regulated *sucC* promoter (sucCup, which is not part of the PhoP regulon) (*Kahramanoglou et al., 2014*) in both the WT and the mutant bacilli. Inset shows a comparable expression of CRP in WT-H37Rv and Δ*phoP*-H37Rv. These results are represented by a schematic model (*Figure 6C*), which suggests that although both mycobacterial regulators are recruited within a subset of CRP-regulated promoters, consistent with CRP-PhoP interaction absence of PhoP strikingly influences in vivo recruitment of CRP.

## Discussion

*M. tuberculosis*, as a successful intracellular pathogen heavily, relies on its ability to sense and appropriately respond to varying environmental conditions over the course of an infection. cAMP, one of the most widely used second messengers, impacts on a wide range of cellular responses in mycobacterial physiology including host-pathogen interactions and virulence (*McDonough and Rodriguez, 2011*). Importantly, cAMP levels are elevated upon infection of macrophages by pathogenic mycobacterium (*Bai et al., 2009*) and addition of cAMP to growing cultures of *M. tuberculosis* significantly impacts mycobacterial gene expression (*Gazdik and McDonough, 2005*). *Agarwal et al., 2009* had shown

**Table 1.** CRP and PhoP binding sites within the commonly regulated promoters[*].

| Rv number | Gene name | PhoP binding sites | Sequence coordinates | CRP binding sites | Sequence coordinates |
|---|---|---|---|---|---|
| Rv0079 | *Rv0079* | CCTCAGCTTCTGCGCAGC | −215 to −232 | GGTGACACAGCCCACA | −95 to −110 |
| Rv0116c | *Rv0116c* | GTACAGCTCGGTCGCAGC | −558 to −576 | TGTGGTCGCGATCACG | +29 to+45 |
| Rv0467 | *icl1* | GAAGAGCGCGGAGCAGATC | +20 to +38 | TGTTACAACGCTCACA | −49 to −64 |
| Rv0469 | *umaA* | GCAAGGCGAGATCACAGA | −88 to −105 | TGTGACAGCCGTTGCG | −326 to −341 |
| Rv1535 | *Rv1535* | GTGGTGCCGAAGCTCTGA | −312 to −330 | GTGGTGCCGAAGCTCT | −314 to −330 |
| Rv2329 | *nark1* | GCTGTTTTCTTGCTGCGA | −194 to −212 | GGTGCGGCAGCCGGCA | +1460 to +1476 |
| Rv2524 | *fas* | GTAGAGCGAATTCCCAGC | −370 to −388 | GATTCCGAGCTGATCGAC | +6014 to +6030 |
| Rv2590 | *fadD9* | TCACAGCCGATCAGCAGC | −104 to −122 | CCCGTGCCGCATCTCAC | −119 to −135 |
| Rv3219 | *whiB1* | CTGGGGTAAGGGCTGTGGA | −191 to −208 | AGTGAGATAGCCCACG | −161 to −176 |
| Rv3616c | *espA* | TCGCAGCGCAGTTGCAGG | −197 to −215 | CGATCAGCACCTCGCG | +2221 to +2237 |

[*]ChIP-sequencing data uncover ~10 promoters that belong to CRP regulon and are also regulated by PhoP. Both CRP and PhoP binding sites were identified in selected *M. tuberculosis* promoters by scanning for respective ChIP-seq data or SELEX-derived consensus sequence motifs (***Galagan et al., 2013***; ***He and Wang, 2014***; ***Kahramanoglou et al., 2014***; ***Solans et al., 2014***). Nucleotide sequences of indicated binding sites are numbered with respect to corresponding ORF start sites.

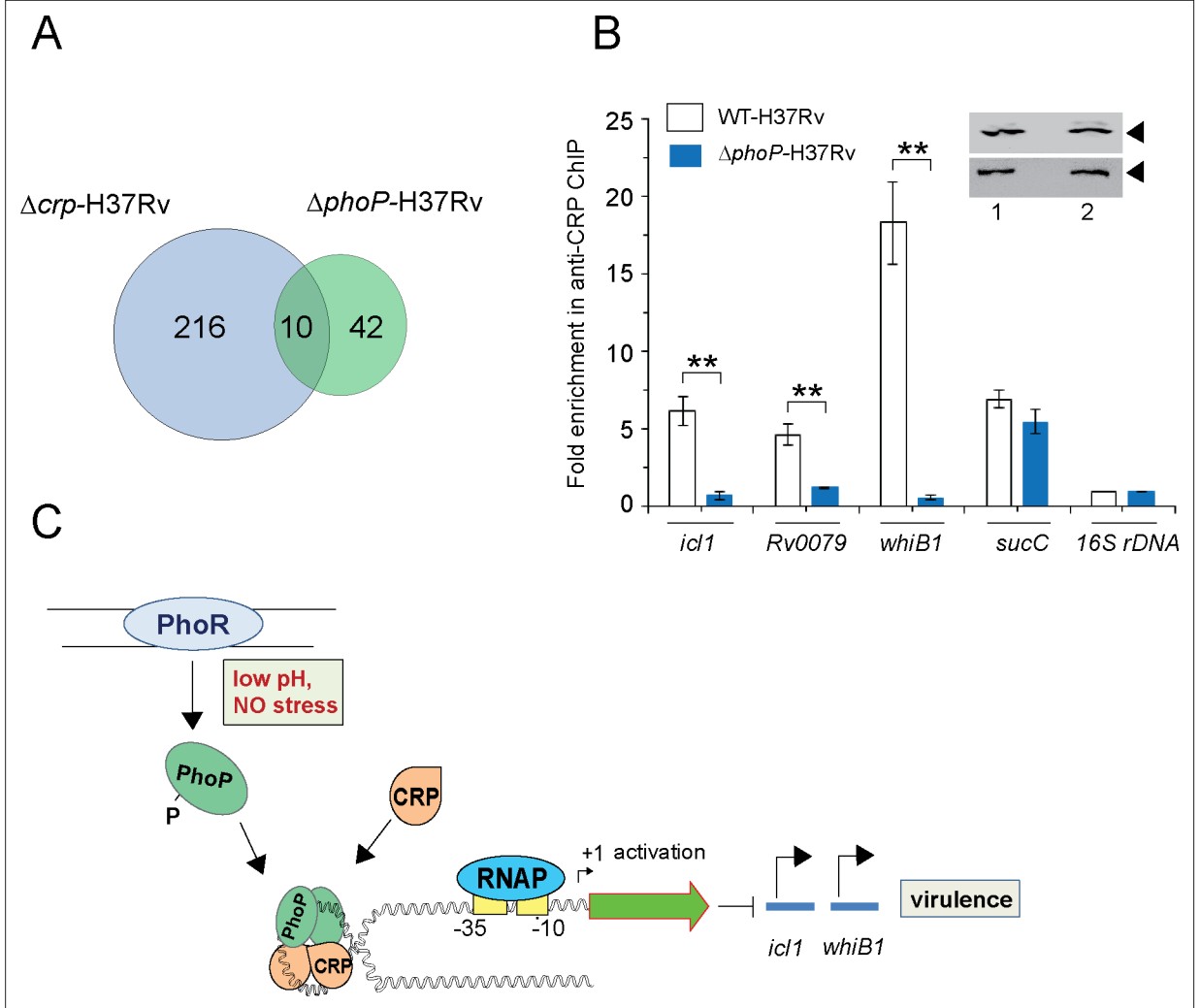

**Figure 6.** PhoP promotes CRP recruitment to regulate *whiB1* expression. (**A**) Venn diagram of genes differentially regulated in *Δcrp*-H37Rv and *ΔphoP*-H37Rv displays an overlap of ~10 promoters. The results are based on previously reported high throughput ChIP-sequencing data of CRP and PhoP (*Kahramanoglou et al., 2014*; *Solans et al., 2014*), respectively. Note that comparisons include genes annotated in *Mycobacterium tuberculosis* H37Rv genome only. (**B**) To examine effect of CRP-PhoP interaction, chromatin-immunoprecipitation (ChIP)-qPCR was carried out using anti-CRP antibody to compare in vivo recruitment of CRP in WT-H37Rv and *ΔphoP*-H37Rv as described in the Methods. Fold PCR enrichment due to CRP binding to indicated promoters was determined by duplicate measurements, each with two technical repeats (**p≤0.01). Inset compares CRP expression in ≈10 µg of indicated crude cell-lysates as probed by anti-CRP antibody; identical extracts were probed with anti-RpoB antibody as a loading control. (**C**) Schematic model showing newly-proposed molecular mechanism of activation of CRP-regulated promoters by simultaneous binding of CRP and PhoP. We propose that the interacting proteins (CRP and PhoP) remain bound to their cognate sites away from the start sites, and stabilize the transcription initiation complex so that RNA polymerase (RNAP) effectively transcribes these genes. Taken together, these molecular events mitigate stress by controlling expression of numerous genes and perhaps contribute to better survival of the bacilli in cellular and animal models.

that bacterial cAMP burst, upon infection of macrophages, improves bacterial survival by interfering with host signaling pathways. However, mechanism of activation of cAMP-responsive/CRP-regulated mycobacterial genes remain poorly understood. This study identified at least part of CRP regulon in *M. tuberculosis*, and established PhoP in addition to CRP and CMR as a third cAMP-responsive transcription factor. Notably, PhoP was required for the regulated expression of a subset of CRP-regulated genes. The CRP-PhoP associated regulon is distinct from the reported CRP regulon (*Bai et al., 2005*; *Kahramanoglou et al., 2014*) with respect to their mechanism of transcription activation, and most likely responds to environmental conditions that regulate its expression.

Having noted that numerous genes regulated by CRP also belong to PhoP regulon and the *whiB1* promoter region displays canonical CRP and PhoP binding sites, we sought to explore a functional link between CRP and PhoP. Importantly, presence of PhoP binding site (*Figure 2B*) was consistent

with in vivo recruitment of PhoP within CRP-regulated promoters (*Figure 1C*). Using in vitro and in vivo approaches, we showed that CRP interacts with PhoP (*Figures 3–4*). While phosphorylation of PhoP was found to be necessary for CRP-PhoP interaction (*Figure 3E*), consistent with cAMP-independent functioning of CRP (*Agarwal et al., 2006*; *Green et al., 2014*; *Stapleton et al., 2010*), mutant CRP proteins defective for cAMP binding showed effective CRP-PhoP interaction (*Figure 5—figure supplement 1C*). To examine how a representative promoter recruits both the regulators in vitro, we attempted an EMSA experiment using purified regulators (*Figure 3A*). The fact that CRP and PhoP together form a ternary complex at the *whiB1* promoter, prompted us to investigate a possible inter-dependence of recruitment of the two regulators. Despite effective recruitment of CRP in WT bacilli, strikingly CRP recruitment within identical promoters was almost undetectable in *ΔphoP*-H37Rv (*Figure 6B*), a result that finds an explanation from CRP-PhoP interaction data (*Figures 3–4*). Given the importance of CRP-regulated essential genes like *whiB1*, and *icl1* in virulence, CRP-PhoP interactions controlling in vivo regulation of expression provide one of the most fundamental biological insights into the mechanism of virulence regulation of *M. tuberculosis*. The finding that PhoP functions as a prerequisite to ensure specific recruitment of CRP within a few CRP-regulated promoters, offers a new biological insight into the controlled expression of CRP regulon. Because such a situation might assist PhoP to function efficiently by ensuring CRP binding only to target promoters already bound to PhoP, it is conceivable that PhoP binding changes DNA conformation, which facilitates CRP recruitment. Therefore, we propose that a DNA-dependent complex interplay of CRP and PhoP via protein-protein contacts most likely controls the precise regulation of these genes. Indeed, this model fits well with, explains and integrates previously reported regulations of *icl1* and *whiB1*, independently by PhoP (*Gonzalo-Asensio et al., 2008*; *Walters et al., 2006*) and CRP (*Kahramanoglou et al., 2014*), respectively.

Based on our observations that the N-domains of PhoP and CRP interact to each other (*Figure 5*), and effective CRP recruitment is possibly ensured within target promoters already bound to PhoP (*Figure 6B*), our model suggests that CRP-PhoP interaction(s) account for activation of CRP-regulated genes. Although we do not provide evidence to suggest 1:1 binding stoichiometry except that as independent regulators, in both cases a dimer of PhoP (*Gupta et al., 2009*; *Menon and Wang, 2011*) and CRP (*Reddy et al., 2009*) bind to target DNA. These considerations take on more significance in the light of previously identified CRP binding site (*Agarwal et al., 2006*; *Stapleton et al., 2010*) and newly identified PhoP binding site (this study) within the *whiB1* promoter. The arrangement and spacing between the two binding sites are suggestive of DNA looping, possibly to facilitate open complex formation and/or assist transcription initiation by contributing additional stability to the initiation complex. Having noted in vivo relevance of CRP-PhoP interaction impacting a few promoters, we sought to explore whether this is a more generalized mechanism involving numerous other CRP-regulated mycobacterial genes. A comparative analysis of genome-wide ChIP-seq data of CRP and PhoP reveals the identity of ~10 promoters (*Table 1*), which appear to be controlled by recruitment of both these regulators (see *Figure 6A*). These results suggest a critically important role of PhoP in binding and transcriptional control of CRP-regulon genes and establish a novel molecular mechanism showing how a complex of two interacting virulence regulators impact the expression of virulence determinants in response to phagosomal stress.

Given the complex life cycle of the intracellular pathogen which encounters a variety of physiological conditions within the human host, an integration of more than one regulator might be required for in vivo functioning to achieve a rather complex regulation of these mycobacterial genes. Notably, the subset of genes that undergo differential expression in *Δcrp*-H37Rv conforms a pattern largely resembling canonical CRP regulon of *E. coli* with CRP binding sites either proximal to transcription start sites, leading to repression or distal to transcription start sites, leading to promoter activation, respectively (*Busby and Ebright, 1999*). It is noteworthy that CRP has been suggested to function as a general chromosomal organizer (*Grainger et al., 2005*). In this study, we uncover that strikingly PhoP binding sites are present next to CRP binding sites, located only distal upstream of promoters, and therefore, associated with activation. We propose that in case of these co-regulated promoters, the additional stability of the transcription initiation complex is derived from protein-protein interaction between CRP and PhoP. These two interacting proteins remain bound to their cognate sites away from the start site, and contribute to stability of the transcription initiation complex, providing access for mycobacterial RNA polymerase (RNAP) to bind and transcribe genes. A schematic model is

shown in *Figure 6C*. Taken together, these molecular events mitigate stress by controlling expression of numerous genes and perhaps contribute to better survival of the bacilli within the host. While this may be possibly a more common strategy than has been previously appreciated, in *M. tuberculosis* essential gene like *whiB1* (*Smith et al., 2010*), virulence determinant like *icl1* (*Gould et al., 2006*; *McKinney et al., 2000*; *Muñoz-Elías and McKinney, 2005*), which remain critical for a successful infection, most likely are under precise regulation as deregulated expression of these genes is likely to alert the immune system to the infection. In conclusion, the present study identifies a new mechanism and provides critically new insights into CRP-dependent gene regulation by the two interacting virulence factors. Future work is required to better understand the components of signaling pathways that make use of this particular mechanism of gene regulation. In addition, the broader questions of whether CRP-dependent mycobacterial gene regulation is triggered by cAMP produced from a specific adenyl cyclase, and whether in response to specific environmental changes, are likely to be exciting areas of future investigation.

## Materials and methods

### Bacterial strains and culture conditions

The strains used were *E. coli* DH5α, for all cloning experiments; *E. coli* BL21 DE3, for protein expression; *M. smegmatis* mc$^2$155, for M-PFC, and reporter-based transcription regulation experiments; WT-H37Rv and Δ*phoP*-H37Rv, in which *phoPR* locus (Rv0757–Rv0758) has been inactivated (*Walters et al., 2006*), for the results reported in this study. *M. tuberculosis* H37Rv, its derivatives and *M. smegmatis* mc$^2$155, were grown at 37°C in Middlebrook 7H9 liquid broth (Difco) containing 0.2% glycerol, 0.05% Tween-80 and 10% ADC (albumin-dextrose-catalase) or on 7H10-agar medium (Difco) containing 0.5% glycerol and 10% OADC (oleic acid-albumin-dextrose-catalase). Growth, transformation of mycobacterial strains and selection of transformants on appropriate antibiotics were carried out as described (*Goyal et al., 2011*). For in vitro growth under specific stress conditions, indicated mycobacterial strains were grown to mid-log phase (OD$_{600}$ 0.4–0.6) and exposed to different stress conditions. For acid stress, cells were initially grown in 7H9 media, pH7.0, and on attaining mid-log phase, it was transferred to acidic media (7H9 media, pH 4.5) for further two hours at 37°C. For oxidative stress, cells were grown in the presence of 50 µM CHP (Sigma-Aldrich) for 24 hr or indicated diamide concentration(s) for 7 days. For NO stress, cells grown to mid-log phase were exposed to 0.5 mM Diethylene triamine NoNoate (DETA NONOate) for 40 min (*Voskuil et al., 2003*).

### Cloning

*M. tuberculosis* full-length *crp* (Rv3676; encoded by 672 bp of the ORF), truncated N-terminal domain *crp*$^N$ (encoded by 267 bp of the ORF), and *crp*$^C$ (encoded by 237 bp of the ORF) over-expressing constructs were cloned in T7-lac-based expression system pET28c (Novagen) between NdeI and HindIII sites as recombinant fusion proteins (containing an N-terminal His$_6$-tag) using primer pairs FPcrp/RPcrp, FPcrp$^N$/RPcrp$^N$, and FPcrp$^C$/RPcrp (*Supplementary file 1a*) resulting in plasmids as listed in *Supplementary file 1b*. Plasmid pGEX-*phoP* expressed GST-PhoP, the full-length PhoP protein with an N-terminal GST-tag (*Gupta et al., 2009*). Likewise, pGEX-*phoP*$^N$ and pGEX-*phoP*$^C$, generated by cloning of corresponding ORFs between BamHI and XhoI sites of pGEX 4T-1 (GE Healthcare), expressed PhoP$^N$, and PhoP$^C$, respectively, each with an N-terminal GST tag. Mutations in CRP were introduced by two-stage overlap extension method using mutagenic primers (*Supplementary file 1a*), and each construct was verified by DNA sequencing.

### Proteins

Full-length and truncated PhoP proteins containing either an N-terminal His$_6$-tag or an N-terminal GST-tag, were purified as described previously (*Gupta et al., 2009*). Wild-type CRP protein (Rv3676) from *M. tuberculosis* H37Rv was expressed in *E. coli* BL21 (DE3) as a fusion protein containing an N-terminal His$_6$-tag (Novagen) and purified by metal-affinity chromatography (Ni-NTA, Qiagen). The protein expression was induced by adding 0.4 mM IPTG in log-phase cultures at O.D$_{600}$ of 0.4, and cells were allowed to grow overnight at 16°C. All subsequent procedures were carried out at 4°C. Briefly, cells were resuspended in lysis buffer (50 mM Tris-HCl, pH 7.9, 500 mM NaCl, 10% glycerol, 0.25% Tween-20, and 50 mM imidazole) followed by addition of lysozyme to a final concentration of

0.1 mg/ml. Next, cell lysates were sonicated, treated with DNaseI for 30 min, and insoluble material was removed by centrifugation for 30 min at 12,000 rpm. The clear supernatant was applied to a column of Ni-NTA agarose (Qiagen) that had been equilibrated with the lysis buffer. After an incubation of 30 min, the column was repetitively washed with 10 column volumes of lysis buffer only, and then with 1 column volume of each of lysis buffer containing 1 M NaCl, lysis buffer containing 50 mM imidazole, and lysis buffer containing 100 mM imidazole, respectively. Finally, the bound protein was eluted in lysis buffer containing 600 mM imidazole. Before storage at –80°C, the protein was extensively dialyzed against buffer A (50 mM Tris-HCl, pH 7.9, 300 mM NaCl, and 10% glycerol), protein concentration was determined using Bradford reagent with BSA as the standard, and expressed in equivalent of protein monomers.

## RNA isolation

Total RNA was extracted from exponentially growing bacterial cultures grown with or without specific stress as described above. Briefly, 25 ml of bacterial culture was grown to mid-log phase ($OD_{600}$=0.4– 0.6) and combined with 40 ml of 5 M guanidinium thiocyanate solution containing 1% β-mercaptoethanol and 0.5% Tween 80. Cells were pelleted by centrifugation, and lysed by resuspending in 1 ml TRIzol (Ambion) in the presence of Lysing Matrix B (100 μm silica beads; MP Bio) using a FastPrep-24 bead beater (MP Bio) at a speed setting of 6.0 for 30 s. The procedure was repeated for 2–3 cycles with incubation on ice in between pulses. Next, cell lysates were centrifuged at 13,000 rpm for 10 min; supernatant was collected and processed for RNA isolation using Direct-Zol RNA isolation kit (ZYMO). Following extraction, RNA was treated with DNAse I (Promega) to degrade contaminating DNA, and integrity was assessed using a Nanodrop (ND-1000, Spectrophotometer). RNA samples were further checked for intactness of 23S and 16S rRNA using formaldehyde-agarose gel electrophoresis, and Qubit fluorometer (Invitrogen).

## Quantitative real-time PCR

cDNA synthesis and PCR reactions were carried out using total RNA extracted from each bacterial culture, and Superscript III platinum-SYBR green one-step qRT-PCR kit (Invitrogen) with appropriate primer pairs (2 μM) using an ABI real-time PCR detection system. Oligonucleotide primer sequences used in RT-qPCR experiments are listed in *Supplementary file 1c*. Control reactions with platinum Taq DNA polymerase (Invitrogen) confirmed the absence of genomic DNA in all our RNA preparations, and endogenously expressed *M. tuberculosis rpoB* was used as an internal control. Fold difference in gene expression was calculated using $\Delta\Delta C_T$ method (*Schmittgen and Livak, 2008*). To determine enrichment due to PhoP and/or CRP binding targets in the IP DNA samples, 1 μl of IP or mock IP (no antibody control) DNA was used with SYBR green (Invitrogen) along with target promoter-specific primers. Average fold differences in mRNA levels were determined from at least two biological repeats each with two technical repeats.

## ChIP-qPCR

ChIP experiments were carried out with some modifications of the protocol described previously (*Bansal et al., 2017*). To determine PhoP recruitment, FLAG tagged *phoP* ORF was expressed in WT-H37Rv and Δ*phoP*-H37Rv from mycobacterial expression vector p19Kpro (*De Smet et al., 1999*), and ChIP was carried out by anti-FLAG antibody. To determine CRP recruitment by ChIP, the CRP-specific antibody was used. *M. tuberculosis* CRP (Rv3676), tagged with $His_6$ at the N-terminus was purified in *E. coli* and used to produce CRP-specific polyclonal antibody in rabbit by AlphaOmegaSciences (India). About 0.3 mg of purified recombinant protein emulsified in Freund's complete adjuvant was administered as primary dose subcutaneously, followed by two booster immunizations in Freund's incomplete adjuvant after 14 and 30 days into New Zealand White Rabbit (~3.8 kg body weight). The titer was determined by the endpoint method, blood collected, serum separated, and stored at –20°C. Total Immunoglobulin G (IgG) was purified from the serum by affinity chromatography using Protein-A resin.

For ChIP assays, mycobacterial cells were grown to mid-exponential phase ($OD_{600}$≈0.4–0.6) and formaldehyde was added to a final concentration of 1%. After incubation of 20 min, glycine was added to a final concentration of 0.5 M to quench the reaction and incubated for further 10 min. Cross-linked cells were harvested by centrifugation and washed two times with ice-cold immunoprecipitation (IP)

buffer (50 mM Tris [pH 7.5], 150 mM NaCl, 1 mM EDTA, 1% Triton X- 100, 1 mM PMSF, and 5% glycerol). Cell pellets were resuspended in 1.0 ml IP buffer containing protease inhibitor cocktail (Roche), cells were lysed and insoluble matter was removed by centrifugation at 13,000 rpm for 10 min at 4°C. Next, supernatant, containing DNA, was sheared to an average size of (add symbol) 500 base pairs (bp) using a Bioruptor (Sonics, VibraCell) with settings of 20 s on and 40 s off for 3–5 min, and split into two aliquots. About 10 µl of sample was analyzed on agarose gel to check the size of the DNA fragments. Each ~0.5 ml supernatant was incubated with either no antibody (mock-IP), or 100 µg specific antibody at 4°C overnight. In a parallel setup, 25 µl Protein A beads were incubated with 50 µg herring sperm DNA in 500 µl of IP buffer on a rotary shaker at 4°C overnight. Next, samples with or without antibody were individually added to tubes containing Protein A beads and incubated at 4°C overnight. Finally, samples were washed two times with IP buffer, two times with IP buffer containing 500 mM NaCl, once with wash buffer (10 mM Tris [pH 8.0], 250 mM LiCl, 1 mM EDTA, 0.5% Tergitol [Sigma-Aldrich], and 0.5% sodium deoxycholate) and once with TE (pH 7.5). IP complexes were then eluted from the resin in 350 µl elution buffer (50 mM Tris [pH 7.5], 10 mM EDTA, 1% SDS, and 100 mM NaHCO$_3$) containing 0.8 mg/ml Proteinase K at 65°C for overnight. The resulting IP samples were then ethanol precipitated.

In vivo recruitment of the regulators were determined using appropriate dilutions of IP DNA in a reaction buffer containing SYBR green mix (Invitrogen), 2 µM PAGE-purified primers (*Supplementary file 1c*), and one unit of Platinum Taq DNA polymerase (Invitrogen). Typically, 40 cycles of amplification were carried out using real-time PCR detection system (Eppendorf). qPCR signal from an IP experiment without adding an antibody (mock) was measured to determine the efficiency of recruitment. In all cases, melting curve analysis confirmed amplification of a single product. Specificity of PCR-enrichment from the identical IP samples was verified using 16S rDNA-specific primers. Each data were collected in duplicate qPCR measurements using at least two bacterial cultures.

## EMSA

The DNA probes were generated by PCR amplification of whiB1up, resolved on agarose gels, recovered by gel extraction, end-labeled with [γ-$^{32}$P ATP] (1000 Ci nmol$^{-1}$) using T4 polynucleotide kinase and purified from free label by Sephadex G-50 spin columns (GE Healthcare). Increasing amounts of purified regulators were incubated with appropriately end-labeled DNA probes in a total volume of 10 µl binding mix (50 mM Tris-HCl, pH 7.5, 50 mM NaCl, 0.2 mg/ml of bovine serum albumin, 10% glycerol, 1 mM dithiothreitol, ≈50 ng of labeled DNA probe, and 0.2 µg of sheared herring sperm DNA) at 20°C for 20 min. DNA-protein complexes were resolved by electrophoresis on a 6% (w/v) polyacrylamide gel (non-denaturing) in 0.5× TBE (89 mM Tris-base, 89 mM boric acid and 2 mM EDTA) at 70 V and 4°C, and radioactive bands were quantified by the phosphorimager (Fuji). To identify composition of retarded complexes, the position of the radioactive material was determined by exposure to a phosphor storage screen, and bands representing the complex(es) were excised from the gel. Next, protein components were extracted from the excised gel fragments in buffer contaning 50 mM Tris, pH 7.5, 50 mM NaCl, 10% glycerol, and 1 mM DTT as described previously (*Anil Kumar et al., 2016*). The eluted protein samples were concentrated, resolved in tricine SDS-PAGE, and detected by Western blotting using appropriate anitbodies.

## Mycobaterial protein fragment complementation (M-PFC) assay

*M. tuberculosis* PhoP was cloned in pUAB400 (kan$^R$; *Supplementary file 1b*) and pUAB400-*phoP* expressed in *M. smegmatis* as described previously (*Singh et al., 2014*). Transformed cells were selected on 7H10/kan plates and grown in liquid medium to obtain competent cells of *M. smegmatis* harboring pUAB400-*phoP*. Likewise, *crp* and *cmr* encoding genes were amplified from *M. tuberculosis* H37Rv genomic DNA using primer pairs FPmCRP/RPmCRP, and FPmCMR/RPmCMR, respectively (*Supplementary file 1a*), and cloned in episomal plasmid pUAB300 (hyg$^R$; *Supplementary file 1b*) between PstI/HindIII and BamHI/HindIII sites, respectively. Each construct was verified by DNA sequencing. Next, co-transformants of *M. smegmatis* were selected on 7H10/kan/hyg plates both in the absence and presence of 10 µg/ml of TRIM to investigate protein-protein interactions (*Singh et al., 2014*). As a positive control, plasmid pair expressing *phoP/phoR* was used as described previously (*Bansal et al., 2017*).

## Acknowledgements

The authors are grateful to G Marcela Rodriguez and Issar Smith (PHRI, New Jersey Medical School - UMDNJ) for ΔphoP-H37Rv, and the complemented *M. tuberculosis* H37Rv strains, Adrie Steyn (University of Alabama) for pUAB300/pUAB400 plasmids, Ashwani Kumar (CSIR-IMTECH) for helpful discussions and Sanjeev Khosla for critical reading of the manuscript. The authors thank Bhuwaneshwar for technical assistance. This study received financial support from intramural grants of CSIR (MLP-0049) and CSIR-IMTECH (OLP-0170), and by a research grant (to D.S) from SERB (EMR/2016/004904), Department of Science and Technology (DST). HK, PP, RRS, and SK were supported by CSIR pre-doctoral fellowships.

## Additional information

### Funding

| Funder | Grant reference number | Author |
| --- | --- | --- |
| CSIR - Institute of Microbial Technology | OLP-0170 | Dibyendu Sarkar |
| Council for Scientific and Industrial Research, India | MLP-0049 | Dibyendu Sarkar |
| Science and Engineering Research Board | EMR/2016/004904 | Dibyendu Sarkar |

The funders had no role in study design, data collection and interpretation, or the decision to submit the work for publication.

### Author contributions

Hina Khan, Data curation, Software, Formal analysis, Validation, Investigation, Project administration; Partha Paul, Conceptualization, Data curation, Software, Formal analysis, Validation; Ritesh Rajesh Sevalkar, Data curation, Investigation, Visualization; Sangita Kachhap, Software, Formal analysis, Investigation; Balvinder Singh, Software, Formal analysis, Investigation, Visualization; Dibyendu Sarkar, Conceptualization, Supervision, Funding acquisition, Investigation, Writing - original draft, Project administration, Writing - review and editing

### Author ORCIDs

Partha Paul ⓘ http://orcid.org/0000-0003-0088-0266
Dibyendu Sarkar ⓘ http://orcid.org/0000-0001-6499-177X

### Decision letter and Author response

Decision letter https://doi.org/10.7554/eLife.80965.sa1
Author response https://doi.org/10.7554/eLife.80965.sa2

## Additional files

### Supplementary files

• Supplementary file 1. Sequence-based nucleic acid reagents. (a) Oligonucleotide primers used for amplification and cloning in this study. (b) Plasmids used in this study. (c) Sequence of oligonucleotide primers used in [a]RT-qPCR and [b]ChIP-qPCR experiments reported in this study.

• Supplementary dataSource data 1. Source data for figures and figure supplements.

• MDAR checklist

### Data availability

All data generated or analysed during this study are included in the manuscript and supporting file. Source data files have been provided for Figures 2-6.

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

# Appendix 1

The supplementary data include four (4) figure supplements, and three supplementary tables (*Supplementary file 1ab and c*, respectively). All data generated or analyzed during this study are included in the manuscript and supporting file; source fata files have been provided for all relevant figures.

## Appendix 1—key resources table

| Reagent type (species) or resource | Designation | Source or reference | Identifiers | Additional information |
|---|---|---|---|---|
| Strain, strain background (*Mycobacterium tuberculosis*) | WT-H37Rv | ATCC25618 | | Wild-type *M. tuberculosis* strain |
| Strain, strain background (*M. tuberculosis*) | ΔphoP-H37Rv | *Walters et al., 2006* | | *phoPR* locus Rv0757-Rv0758 has been inactivated |
| Strain, strain background (*M. tuberculosis*) | ΔphoP::phoP | *Walters et al., 2006* | | ΔphoP-complemented with *phoP* |
| Strain, strain background (*M. tuberculosis*) | ΔphoP:: phoPD71N | This study | | ΔphoP-complemented with *phoPD71N* |
| Strain, strain background (*M. smegmatis*) | M. smegmatis mc$^2$155 | *Snapper et al., 1990* | | |
| Sequence-based reagent | FPwhiB1$^{up}$ | This study | PCR primers | whiB1$^{up}$ forward primer: AATAATAAGCTTAGATGTGATGGG |
| Sequence-based reagent | RPwhiB1$^{up}$ | This study | PCR primers | whiB1$^{up}$ reverse primer: AATAATGGTACCTACCGGGAAGAA |
| Sequence-based reagent | FPwhiB1$^{upmut}$ | This study | PCR primers | mutagenic forward primer for whiB1$^{upmut}$: GTCAAACAAGGATCACAAAACGAGATCGCCA |
| Sequence-based reagent | RPwhiB1$^{upmut}$ | This study | PCR primers | mutagenic reverse primer for whiB1$^{upmut}$: TTGTGATCCTTGTTTGACTGTACTACACTA |
| Sequence-based reagent | FPwhiB1$^{up1}$ | This study | PCR primers | whiB1$^{up1}$ forward primer: AATAATAAGCTTAGATGTGATGGG |
| Sequence-based reagent | RPwhiB1$^{up1}$ | This study | PCR primers | whiB1$^{up1}$ reverse primer: AATAATGGTACCAGCGCGTGGGCT |
| Sequence-based reagent | FPwhiB1$^{up2}$ | This study | PCR primers | whiB1$^{up2}$ forward primer: AATAATGGATCCTACGTAACACTA |
| Sequence-based reagent | RPwhiB1$^{up2}$ | This study | PCR primers | whiB1$^{up2}$ reverse primer: AATAATGGTACCAGCGCGTGGGCT |
| Sequence-based reagent | FPwhiB1$^{up3}$ | This study | PCR primers | whiB1$^{up3}$ forward primer: AATAATTCCGAAGAAACGCCT |
| Sequence-based reagent | FPphoPstart | This study | PCR primers | His-tagged *phoP* forward primer: AATAATGGATCCATGCGGAAAGGGGT |
| Sequence-based reagent | RPphoPFLAG | This study | PCR primers | FLAG-tagged *phoP* reverse primer: AATAAGCTTTCACTTGTCGTCAT CGTCTTTGTAGTCTCGAGGCTCCCGCA |
| Sequence-based reagent | FPmCMR | This study | PCR primers | *cmr* forward primer: AATAATTGTTCGACTGACG |
| Sequence-based reagent | RPmCMR | This study | PCR primers | *cmr* reverse primer: AATAATCGGGTTTGTGTTGT |
| Sequence-based reagent | FPmCRP | This study | PCR primers | *crp* forward primer: AATAATAAGCTTGTGGACGAGATC |
| Sequence-based reagent | RPmCRP | This study | PCR primers | *crp* reverse primer: AATAATATCGATTTACCTCGCTCG |
| Sequence-based reagent | FPPhoP$^N$ | *Pathak et al., 2010* | PCR primers | phop$^N$ forward primer: CCTGGATCCATGCGGAAAGGGGTT |
| Sequence-based reagent | RPPhoP$^N$ | This study | PCR primers | phoP$^N$ reverse primer: AATAATCTCGAGGCGTCGCAGGATGA |
| Sequence-based reagent | FPPhoP$^C$ | This study | PCR primers | phop$^C$ forward primer: AATAATCGGATCCAAGGGCAACAAGGAACCA |
| Sequence-based reagent | RPPhoP$^C$ | *Pathak et al., 2010* | PCR primers | phoP$^C$ reverse primer: GGTCTCGAGTCGAGGCTCCCGCAG |

*Appendix 1 Continued on next page*

*Appendix 1 Continued*

| Reagent type (species) or resource | Designation | Source or reference | Identifiers | Additional information |
|---|---|---|---|---|
| Sequence-based reagent | FPCRP | This study | PCR primers | *crp* forward primer: AATAATCATATGGTGGACGAGATC |
| Sequence-based reagent | FPCRP$^N$ | This study | PCR primers | *crp*$^N$ forward primer: ATATATGGATCCCCCGTCGACTTCCCCC |
| Sequence-based reagent | RPCRP$^N$ | This study | PCR primers | *crp*$^N$ reverse primer: ATATATAAGCTTTCAACGATCGGCGAT |
| Sequence-based reagent | FPCRP$^C$ | This study | PCR primers | *crp*$^C$ forward primer: ATATATCATATGGTGCCCGGTCGGGT |
| Sequence-based reagent | RPCRP | This study | PCR primers | *crp* reverse primer: AATAATCTCGAGTTACCTCGCTCG |
| Sequence-based reagent | FPCRPG79A | This study | PCR primers | mutagenic *crp* forward primer: ATGTTCGCGGAGTTGTCGATCT |
| Sequence-based reagent | RPCRPG79A | This study | PCR primers | mutagenic *crp* reverse primer: CGACAACTCCGCGAACATGTCCGA |
| Sequence-based reagent | FPCRPT90A | This study | PCR primers | mutagenic *crp* forward primer: GTCCGCGCGCGTCCAGC |
| Sequence-based reagent | RPCRPT90A | This study | PCR primers | mutagenic *crp* reverse primer: GCTGGACGCGCGCGGACCCGGGT |
| Sequence-based reagent | FPPhoP62-64A | This study | PCR primers | mutagenic *phoP* forward primer: GCAGCAGCACCGGACGCGGTG |
| Sequence-based reagent | RPPhoP62-64A | This study | PCR primers | mutagenic *phoP* reverse primer: TGCTGCTGCCCGGGCCCGATC |
| Sequence-based reagent | FPPhoP76-78A | This study | PCR primers | mutagenic *phoP* forward primer: GCAGCAGCAGGCTTTGGGGTG |
| Sequence-based reagent | RPPhoP76-78A | This study | PCR primers | mutagenic *phoP* reverse primer: TGCTGCTGCGGGCATCATCAC |
| Sequence-based reagent | FPPhoP105-107A | This study | PCR primers | mutagenic *phoP* forward primer: GCAGCAGCAATCGCGGGTCTG |
| Sequence-based reagent | RPPhoP105-107A | This study | PCR primers | mutagenic *phoP* reverse primer: TGCTGCTGCTAGCGAGTCACG |
| Sequence-based reagent | FPPhoP110-112A | This study | PCR primers | mutagenic *phoP* forward primer: GCAGCAGCACTGGGTGGTGAC |
| Sequence-based reagent | RPPhoP110-112A | This study | PCR primers | mutagenic *phoP* reverse primer: TGCTGCTGCCGCGATCTTGTC |
| Sequence-based reagent | FPPhoP118-120A | This study | PCR primers | mutagenic *phoP* forward primer: GCAGCAGCAAAGCCCTTCAGT |
| Sequence-based reagent | RPPhoP118-120A | This study | PCR primers | mutagenic *phoP* reverse primer: TGCTGCTGCGTCGTCACCACC |
| Sequence-based reagent | FPaprART | **Bansal et al., 2017** | RT-qPCR primers | gene specific primer: TTGACCATGACAGCGAGTGT |
| Sequence-based reagent | RPaprART | **Bansal et al., 2017** | RT-qPCR primers | gene specific primer: TTGGACAGAAATGCAGGATG |
| Sequence-based reagent | FPcrpRT | This study | RT-qPCR primers | gene specific primer: ATCATCATCTCGGGGAAGGT |
| Sequence-based reagent | RPcrpRT | This study | RT-qPCR primers | gene specific primer: CAGCTGTTCGGAGATTTCG |
| Sequence-based reagent | FPcmrRT | This study | RT-qPCR primers | gene specific primer: ATTGGCCGAAACGTTACAAG |
| Sequence-based reagent | RPcmrRT | This study | RT-qPCR primers | gene specific primer: ACCATCGGCATCTCCAGTAG |
| Sequence-based reagent | FPicl1RT | This study | RT-qPCR primers | gene specific primer: GCTTCTACCGCACCAAGAAC |
| Sequence-based reagent | RPicl1RT | This study | RT-qPCR primers | gene specific primer: TCGAGGTGCTTTTTCCAGTT |
| Sequence-based reagent | FPphoPRT | This study | RT-qPCR primers | gene specific primer: GCCTCAAGTTCCAGGGCTTT |

*Appendix 1 Continued on next page*

*Appendix 1 Continued*

| Reagent type (species) or resource | Designation | Source or reference | Identifiers | Additional information |
|---|---|---|---|---|
| Sequence-based reagent | RPphoPRT | This study | RT-qPCR primers | gene specific primer: CCGGGCCCGATCCA |
| Sequence-based reagent | FPumaART | This study | RT-qPCR primers | gene specific primer: CGTTATGCGGCATTCTTTG |
| Sequence-based reagent | RPumaART | This study | RT-qPCR primers | gene specific primer: TGCGCAAATTTGAAGATGTC |
| Sequence-based reagent | FPwhiB1RT | This study | RT-qPCR primers | gene specific primer: CACAAGGCGGTCTGTCGT |
| Sequence-based reagent | RPwhiB1RT | This study | RT-qPCR primers | gene specific primer: GAGTCCTGGCCGGTATTCAG |
| Sequence-based reagent | FPrpoBRT | This study | RT-qPCR primers | gene specific primer: GGAGGCGATCACACCGCAGACGTT |
| Sequence-based reagent | RPrpoBRT | This study | RT-qPCR primers | gene specific primer: CCTCCAGCCCGGCACGCTCACGT |
| Sequence-based reagent | FP16SrDNA RT | This study | RT-qPCR primers | gene specific primer: CTGAGATACGGCCCAGACTC |
| Sequence-based reagent | RP16SrDNA RT | This study | RT-qPCR primers | gene specific primer: CGTCGATGGTGAAAGAGGTT |
| Sequence-based reagent | FPespA[up] | **Anil Kumar et al., 2016** | ChIP-qPCR primers | promoter specific primer: CGTGATCTTGATACGGCTCG |
| Sequence-based reagent | RPespA[up] | **Anil Kumar et al., 2016** | ChIP-qPCR primers | promoter specific primer: GTTGTTGGTACCCTCGGCAAGATCGGC |
| Sequence-based reagent | FPgapdh[up] | This study | ChIP-qPCR primers | promoter specific primer: GAGTAGGCATCAACGGGTTTG |
| Sequence-based reagent | RPgapdh[up] | This study | ChIP-qPCR primers | promoter specific primer: GTGCTGTTGTCGGTGATGTC |
| Sequence-based reagent | FPicl1[up] | This study | ChIP-qPCR primers | promoter specific primer: AATAATAAGCTTACCGGATCCGCA |
| Sequence-based reagent | RPicl1[up] | This study | ChIP-qPCR primers | promoter specific primer: AATAATGGTACCGTTCGTGTCC |
| Sequence-based reagent | FP16SrDNA[up] | **Singh et al., 2014** | ChIP-qPCR primers | promoter specific primer: CTGAGATACGGCCCAGACTC |
| Sequence-based reagent | RP16SrDNA[up] | **Singh et al., 2014** | ChIP-qPCR primers | promoter specific primer: CGTCGATGGTGAAAGAGGTT |
| Sequence-based reagent | FPsucC[up] | This study | ChIP-qPCR primers | promoter specific primer: GGCTGTGATTGTGAGTTGGA |
| Sequence-based reagent | RPsucC[up] | This study | ChIP-qPCR primers | promoter specific primer: GCGAATAACTCCTTGGCTTG |
| Sequence-based reagent | FPumaA[up] | This study | ChIP-qPCR primers | promoter specific primer: TGTTGCTGCGTATGGTTGAG |
| Sequence-based reagent | RPumaA[up] | This study | ChIP-qPCR primers | promoter specific primer: AATCGATTGCGACTCTTCGT |
| Sequence-based reagent | FPwhiB1[up1] | This study | ChIP-qPCR primers | promoter specific primer: AATAATAAGCTTAGATGTGATGGG |
| Sequence-based reagent | RPwhiB1[up1] | This study | ChIP-qPCR primers | promoter specific primer: AATAATGGTACCAGCGCGTGGGCT |
| Recombinant DNA reagent | pET-*phoP* | **Gupta et al., 2009** | Plasmid DNA | His$_6$ tagged-PhoP residues 1–247 cloned in pET15b |
| Recombinant DNA reagent | pGEX-*phoP* | **Gupta et al., 2009** | Plasmid DNA | PhoP residues 1–247 cloned in pGEX-4T-1 |
| Recombinant DNA reagent | pGEX-*phoP*LAla5 | This study | Plasmid DNA | G142-P146 residues mutated to A in *phoP* of pGEX-*phoP* |
| Recombinant DNA reagent | pGEX-*phoP*[N] | This study | Plasmid DNA | PhoP residues 1–141 cloned in pGEX-4T-1 |
| Recombinant DNA reagent | pGEX-*phoP*[C] | This study | Plasmid DNA | PhoP residues 141–247 cloned in pGEX-4T-1 |
| Recombinant DNA reagent | pGEX-*phoP*(62-64)Ala | This study | Plasmid DNA | E62-R64 residues mutated to A in *phoP* of pGEX-*phoP* |

*Appendix 1 Continued on next page*

*Appendix 1 Continued*

| Reagent type (species) or resource | Designation | Source or reference | Identifiers | Additional information |
|---|---|---|---|---|
| Recombinant DNA reagent | pGEX-*phoP*(76-78)Ala | This study | Plasmid DNA | G76-D78 residues mutated to A in *phoP* of pGEX-*phoP* |
| Recombinant DNA reagent | pGEX-*phoP*(105-107)Ala | This study | Plasmid DNA | Q105-K107 residues mutated to A in *phoP* of pGEX-*phoP* |
| Recombinant DNA reagent | pGEX-*phoP*(110-112)Ala | This study | Plasmid DNA | G110-T112 residues mutated to A in *phoP* of pGEX-*phoP* |
| Recombinant DNA reagent | pGEX-*phoP*(118-120)Ala | This study | Plasmid DNA | Y118-T120 residues mutated to A in *phoP* of pGEX-*phoP* |
| Recombinant DNA reagent | pME1mL1-*phoP*[b] | *Goyal et al., 2011* | Plasmid DNA | PhoP residues 1–247 cloned in pME1mL1 |
| Recombinant DNA reagent | pSM128[c] | *Dussurget et al., 1999* | Plasmid DNA | Integrative promoter probe vector for mycobacteria |
| Recombinant DNA reagent | pSM-*whiB1*[up] | This study | Plasmid DNA | whiB1[up]-*lacZ* fusion in pSM128 |
| Recombinant DNA reagent | pSM-*whiB1*[upmut] | This study | Plasmid DNA | pSM-*whiB1*[up] carrying changes in the PhoP binding site |
| Recombinant DNA reagent | pUAB400[d] | *Singh et al., 2006* | Plasmid DNA | Integrative mycobacteria - *E. coli* shuttle plasmid, Kan[r] |
| Recombinant DNA reagent | pUAB400-*phoP* | *Singh et al., 2014* | Plasmid DNA | PhoP residues 1–247 cloned in pUAB400 |
| Recombinant DNA reagent | pUAB300[b] | *Singh et al., 2006* | Plasmid DNA | Episomal mycobacteria - *E. coli* shuttle plasmid, Hyg[r] |
| Recombinant DNA reagent | pUAB300-*crp* | This study | Plasmid DNA | CRP residues 1–224 cloned in pUAB300 |
| Recombinant DNA reagent | pUAB300-*cmr* | This study | Plasmid DNA | CMR residues 1–244 cloned in pUAB300 |
| Recombinant DNA reagent | pET-28c[d] | Novagen | Plasmid DNA | *E. coli* cloning vector, Kan[r] |
| Recombinant DNA reagent | pET-*crp*[d] | This study | Plasmid DNA | His$_6$ tagged-CRP residues 1–224 cloned in pET-28c |
| Recombinant DNA reagent | pET-*crp*[N] | This study | Plasmid DNA | His$_6$ tagged -CRP residues 28–116 cloned in pET-28c |
| Recombinant DNA reagent | pET-*crp*[C] | This study | Plasmid DNA | His$_6$ tagged-CRP residues 146–224 cloned in pET-28c |
| Recombinant DNA reagent | p19Kpro[b] | *De Smet et al., 1999* | Plasmid DNA | Mycobacteria expression vector |
| Recombinant DNA reagent | p19Kpro-*phoP* | *Anil Kumar et al., 2016* | Plasmid DNA | His$_6$-tagged PhoP residues 1–247 cloned in p19Kpro |
| Recombinant DNA reagent | p19Kpro-*phoP*[FLAG] | This study | Plasmid DNA | FLAG-tagged PhoP residues 1–247 cloned in p19Kpro |
| Antibody | Anti-CRP (rabbit polyclonal) | AlphaOmega Sciences (This study) | | 1:5000 |
| Antibody | Anti-FLAG (rabbit polyclonal) | Invitrogen | #PA1-984B | 1:3000 |
| Antibody | Anti-GST (goat polyclonal) | GE Healthcare | # 27457701V | 1:5000 |
| Antibody | Anti-His (mouse monoclonal) | Invitrogen | #MA1-21315 | 1:5000 |
| Antibody | Anti-PhoP (rabbit polyclonal) | AlphaOmegaSciences (This study) | | 1:3000 |

