## [Editor Report]

This paper will be of broad interest to those working on the regulation of gene expression and mycobacteria. The article presents data to elucidate the collaboration between two important transcriptional regulators to selectively turn on the expression of a gene set in *Mycobacterium tuberculosis*.

---

## [Decision Letter]

**Decision letter after peer review:**

Thank you for submitting your article "Convergence of two global regulators to coordinate expression of essential virulence determinants of *Mycobacterium tuberculosis*" for consideration by *eLife*. Your article has been reviewed by 2 peer reviewers, and the evaluation has been overseen by a Reviewing Editor and Bavesh Kana as the Senior Editor. The following individual involved in the review of your submission has agreed to reveal their identity: Valakunja Nagaraja (Reviewer #1).

Essential revisions:

1) Data Interpretations need to be revisited so that they are not overstated.

2) The organization and presentation need to be more appropriate for the broad audience of *eLife*.

3) See below for necessary controls for the experiments.

*Reviewer #1 (Recommendations for the authors):*

1. The authors may discuss how CRP and PhoP interaction activates transcription.

2. The complex in addition to increasing /stabilizing the closed complex, may be required for open complex formation and or stability.

3. Although modeling experiments reveal the interaction between the two proteins is cAMP-independent, and authors justify the non-inclusion of vAMP in the interaction assays, it is conceivable that cAMP binding may have an effect due to conformational changes in CRP triggered by cAMP binding.

4. The studies presented do not unequivocally establish whether the two regulators interact on their own in absence of DNA. The data also shows that DNA-bound PhoP recruits CRP. The protein-protein interaction assays do not rule out the presence of DNA in these assays. DNA is notorious for contamination! A control would be DNAse treated cell lysates in such assays.

5. The discussion (also in abstract) needs to be rewritten in parts with a more precise interpretation of the results. For example, they state that they have uncovered a new mechanism of transcription activation. What is shown is the need for a complex of activators to turn on gene expression.

6. Figure 2B versus Figure 3A – in the text it stated that robust binding in the former and modest binding in the latter.

*Reviewer #2 (Recommendations for the authors):*

1. My greatest concern is that, in its current state, the manuscript is very difficult to follow. I recommend thorough editing and reorganization that would make the manuscript more accessible to all, especially those outside of the authors' immediate field. I have outlined areas that could be improved, point by point, below:

– There is no introductory material on the PhoP signal transduction system (the star of the show) which makes the rest of the manuscript very difficult to follow. For example, PhoP introduction would be of use to explain why the authors assess NO and acid stress in Figure 1A, which otherwise comes out of nowhere. Though the biology of PhoP may be of second nature to the authors, I know little about it (what is its cognate receptor, what are the inputs to the system, what does PhoP regulate, etc, etc). Notably, however, there is a comprehensive introduction to the cAMP-CRP system, which is of much less emphasis in the actual manuscript that follows.

– There are many acronyms that are not defined. I do not know what ∆phoP-H37Rv is, I do not know what TRIM is, I do not know what M-PFC is, Atc, etc… Every acronym should be introduced at first use. Along these lines, I do not think it is necessary, in the Results section, to provide plasmid and strain numbers, unless essential. That you used pUAB300, for example, is of little consequence to the reader and is distracting. This information should be contained in the methods section.

– The labeling and naming conventions used by the authors are confusing. Referring to the N- or C-terminal truncated PhoP proteins, the authors use PhoPN and PhoPC. This naming convention is reminiscent of a gene pair, such as a two-component system. N and C can be superscripted to help the reader. Likewise whiB1up1, etc could also be superscripted, and so on.

– Overall, the results and figures could be streamlined to make the manuscript more straightforward, shortened, and easier to follow. In numerous places, the authors present data that are repetitive with their own findings (often in the same figure, for example, Figure 2A and 2B are similar, and Figure 2D and 2E corroborate). In these cases, the follow-up results can be presented in the supplement (Figure 2B and Figure 2E for example), and little text is needed to describe them (To confirm…). Moreover, in some figures, a good deal of negative data is shared (for example, Figure 5A, B, E) and these panels can be moved to the supplement and described in concise language. These instances distract from the main message of the manuscript.

– The manuscript contains numerous grammatical and syntax errors. Thorough proofreading is needed. (e.g. the final sentence of the abstract does not make sense).

2. The authors state that CRP and PhoP regulate some genes individually and some in common. This needs to be displayed as a figure and discussed. How many genes are regulated individually and in common, what are they?

3. Why does whiB1 not complement Figure 1A? The authors glance over this, though it is a critical experiment as whiB1 is the major focus of the remainder of the manuscript.

4. The authors should use phospho-dead phoPD71N throughout the manuscript. It would be compelling to see results for a phoPD71N in vivo (for example as in Figure 1A), and with purified protein, for example in EMSAs, where one would predict that no shift would occur if the authors' hypothesis is correct. As a side note, the authors should provide the citation showing that this mutant is incapable of phosphorylation.

5. I do not see what the computational docking of CRP and PhoP adds to the manuscript. If these results provided you with interfacing residues that could be used to guide point mutagenesis or future experiments, then it would be worth including, however, it does not do so in this case. Removing this section would be another way to shorten/streamline the manuscript.

---

## [Author Response]

Reviewer #1 (Recommendations for the authors):1. The authors may discuss how CRP and PhoP interaction activates transcription.

We have now incorporated new text discussing how CRP-PhoP interaction activates transcription in the ‘Discussion’ section of the revised manuscript. Please see our response to the previous comment.

2. The complex in addition to increasing /stabilizing the closed complex, may be required for open complex formation and or stability.

We totally agree. We have now mentioned it in the 3^rd^ paragraph of the ‘Discussion’ section. The following text was included in the revised manuscript.

“The arrangement and spacing between the two binding sites are suggestive of DNA looping, possibly to facilitate open complex formation and/or assist transcription initiation by contributing additional stability to the initiation complex.”

3. Although modeling experiments reveal the interaction between the two proteins is cAMP-independent, and authors justify the non-inclusion of vAMP in the interaction assays, it is conceivable that cAMP binding may have an effect due to conformational changes in CRP triggered by cAMP binding.

Following the reviewer’s suggestion, we have now incorporated text in the ‘Results’ section of the revised manuscript (please see below). Note that the other reviewer requested that we remove the structural docking results.

“Although it is conceivable that cAMP binding may induce a conformational change of CRP, in keeping with cAMP-independent functioning of *M. tuberculosis* CRP (Agarwal et al., 2006; Green et al., 2014; Stapleton et al., 2010), the mutant proteins over a range of concentrations showed effective DNA binding to end-labelled whiB1^up^ with a comparable affinity as that of the wild-type CRP (Figure 5—figure supplement 1D).”

4. The studies presented do not unequivocally establish whether the two regulators interact on their own in absence of DNA. The data also shows that DNA-bound PhoP recruits CRP. The protein-protein interaction assays do not rule out the presence of DNA in these assays. DNA is notorious for contamination! A control would be DNAse treated cell lysates in such assays.

The reviewer raises an interesting point. We have now performed both in vitro and in vivo CRP-PhoP interaction experiments using DNase I -treated cell extracts. The figures of the previous version (Figures 4A-B) are now replaced with new figures (also Figures 4A-B), and figure captions with relevant results have been adjusted accordingly in the revised manuscript.

Also, DNaseI treatment of cell extracts have been mentioned in the respective figure captions of the revised manuscript.

5. The discussion (also in abstract) needs to be rewritten in parts with a more precise interpretation of the results. For example, they state that they have uncovered a new mechanism of transcription activation. What is shown is the need for a complex of activators to turn on gene expression.

Following reviewer’s recommendations, we have made numerous changes throughout the ‘Discussions’ section for a more precise interpretation of the results. Also, part of the text on cAMP-CRP system, which is less relevant in this study, has been deleted in the revised manuscript.

We also thank the reviewer for a specific suggestion. Accordingly, we have made changes both in the ‘Abstract’ as well as ‘Discussion’ sections of the revised manuscript.

6. Figure 2B versus Figure 3A – in the text it stated that robust binding in the former and modest binding in the latter.

We apologize for the inaccuracy that led to this confusion. We have now replaced ‘modest’ with ‘effective’ in the ‘Results’ section describing results related to Figure 3A in the revised manuscript.

Reviewer #2 (Recommendations for the authors):1. My greatest concern is that, in its current state, the manuscript is very difficult to follow. I recommend thorough editing and reorganization that would make the manuscript more accessible to all, especially those outside of the authors' immediate field. I have outlined areas that could be improved, point by point, below:

Following reviewer’s suggestions, we have now performed thorough editing and reorganization of the manuscript. To this effect, numerous changes were made throughout the manuscript (please see below).

– There is no introductory material on the PhoP signal transduction system (the star of the show) which makes the rest of the manuscript very difficult to follow. For example, PhoP introduction would be of use to explain why the authors assess NO and acid stress in Figure 1A, which otherwise comes out of nowhere. Though the biology of PhoP may be of second nature to the authors, I know little about it (what is its cognate receptor, what are the inputs to the system, what does PhoP regulate, etc, etc). Notably, however, there is a comprehensive introduction to the cAMP-CRP system, which is of much less emphasis in the actual manuscript that follows.

We have now incorporated new text describing biology of PhoP in the third paragraph of Introduction section (revised manuscript). Also, part of the text on cAMP-CRP system, which is less relevant in this study, has been deleted from the ‘Introduction’ section of the revised manuscript.

– There are many acronyms that are not defined. I do not know what ∆phoP-H37Rv is, I do not know what TRIM is, I do not know what M-PFC is, Atc, etc… Every acronym should be introduced at first use. Along these lines, I do not think it is necessary, in the Results section, to provide plasmid and strain numbers, unless essential. That you used pUAB300, for example, is of little consequence to the reader and is distracting. This information should be contained in the methods section.

Following reviewer’s recommendation, we have now gone through the manuscript and made the suggested changes throughout (please see below).

(i) We have introduced each acronym at the first use in the text.

(ii) We have now removed plasmid/strain numbers from the ‘Results’ section of the main text, and made sure to restrict them only in the methods section.

– The labeling and naming conventions used by the authors are confusing. Referring to the N- or C-terminal truncated PhoP proteins, the authors use PhoPN and PhoPC. This naming convention is reminiscent of a gene pair, such as a two-component system. N and C can be superscripted to help the reader. Likewise whiB1up1, etc could also be superscripted, and so on.

We are sorry for the confusing labelling. Following reviewer’s recommendations, we have now modified labelling of domain constructs, upstream regulatory regions etc. throughout the manuscript including figures and figure captions.

– Overall, the results and figures could be streamlined to make the manuscript more straightforward, shortened, and easier to follow. In numerous places, the authors present data that are repetitive with their own findings (often in the same figure, for example, Figure 2A and 2B are similar, and Figure 2D and 2E corroborate). In these cases, the follow-up results can be presented in the supplement (Figure 2B and Figure 2E for example), and little text is needed to describe them (To confirm…). Moreover, in some figures, a good deal of negative data is shared (for example, Figure 5A, B, E) and these panels can be moved to the supplement and described in concise language. These instances distract from the main message of the manuscript.

Following reviewer’s recommendations, we have now made numerous changes in the revised manuscript. These are as follows:

(A) To eliminate repetition, Figures 2B and 2E of the original version have been moved to the supplemental data [Figure 2—figure supplement 1(B-C)] of the revised manuscript.

(B) The results related to both the main data and follow-up experiments have been shortened, and presented in a more concise way.

(C) Negative data of Figure 5 (Figures 5A, 5B and 5E) have been moved to the supplement [Figure 5—figure supplement 1(A-C), revised manuscript], and the results described concisely.

(D) Also, we have thoroughly proof-read ‘results’ section to shorten text for a more precise and straightforward presentation of the data.

– The manuscript contains numerous grammatical and syntax errors. Thorough proofreading is needed. (e.g. the final sentence of the abstract does not make sense).

We are sorry for the inaccuracies. We have done thorough editing of the manuscript and made numerous corrections throughout. Also, the final sentence of the abstract has been modified.

2. The authors state that CRP and PhoP regulate some genes individually and some in common. This needs to be displayed as a figure and discussed. How many genes are regulated individually and in common, what are they?

The reviewer raises an interesting question. As recommended, we have now included a Venn diagram showing commonly and independently regulated genes by the mycobacterial regulators CRP and PhoP (see Figure 6A). The new figure, figure caption and the related results have been adjusted accordingly in the revised manuscript.

3. Why does whiB1 not complement Figure 1A? The authors glance over this, though it is a critical experiment as whiB1 is the major focus of the remainder of the manuscript.

The reviewer raises an important question. In this study, we have compared *phoP* expression in WT-H37Rv and the complemented mutant strain (*∆phoP::phoP*) [Figure 1—figure supplement 1(A-B)]. Our results clearly demonstrate that *phoP* expression level is reproducibly higher in the complemented mutant relative to the WT-H37Rv. These results account for an elevated mRNA levels of *icl1* and *umaA* in the complemented mutant relative to WT-H37Rv. However, relatively poor restoration of *whiB1* expression in the complemented mutant (relative to WT-H37Rv) is possibly related to inadequate activation of PhoPR in the complemented mutant.

To address reviewer’s concern, we have now included the above text in the ‘Results’ section of the revised manuscript.

4. The authors should use phospho-dead phoPD71N throughout the manuscript. It would be compelling to see results for a phoPD71N in vivo (for example as in Figure 1A), and with purified protein, for example in EMSAs, where one would predict that no shift would occur if the authors' hypothesis is correct. As a side note, the authors should provide the citation showing that this mutant is incapable of phosphorylation.

As recommended by the reviewer, the following changes were made.

(a) We have included new results with PhoPD71N in the regulation experiment described in Figure 1A. The Results section and the figure caption have been adjusted accordingly in the revised manuscript.

(b) We have now included EMSA data using purified PhoPD71N in the revised manuscript (Figure 2D). The new figure, figure caption and the relevant Results section have been adjusted accordingly.

(c) in vitro pull-down assay studying interaction between CRP and PhoPD71N has now been included as a part of Figure 4 (Figure 4C, revised version).

(d) Also, reference citing PhoPD71N, a mutant PhoP protein incapable of phosphorylation at Asp-71 has been included in the ‘Results’ section (page 6) of the revised manuscript.

5. I do not see what the computational docking of CRP and PhoP adds to the manuscript. If these results provided you with interfacing residues that could be used to guide point mutagenesis or future experiments, then it would be worth including, however, it does not do so in this case. Removing this section would be another way to shorten/streamline the manuscript.

We have now removed the data related to structural docking of CRP and PhoP from the revised manuscript.